# Experimental quantum simulation of superradiant phase transition beyond no-go theorem via antisqueezing

Xi Chen[1,2,3,5], Ze Wu[1,2,3,5], Min Jiang[1,2,3], Xin-You Lü[4 ✉], Xinhua Peng [1,2,3 ✉] & Jiangfeng Du [1,2,3]

The superradiant phase transition in thermal equilibrium is a fundamental concept bridging statistical physics and electrodynamics, which has never been observed in real physical systems since the first proposal in the 1970s. The existence of this phase transition in cavity quantum electrodynamics systems is still subject of ongoing debates due to the no-go theorem induced by the so-called $A^2$ term. Moreover, experimental conditions to study this phase transition are hard to achieve with current accessible technology. Based on the platform of nuclear magnetic resonance, here we experimentally simulate the occurrence of an equilibrium superradiant phase transition beyond no-go theorem by introducing the antisqueezing effect. The mechanism relies on that the antisqueezing effect recovers the singularity of the ground state via exponentially enhancing the zero point fluctuation of system. The strongly entangled and squeezed Schrödinger cat states of spins are achieved experimentally in the superradiant phase, which may play an important role in fundamental tests of quantum theory and implementations of quantum metrology.

[1] Hefei National Laboratory for Physical Sciences at the Microscale and Department of Modern Physics, University of Science and Technology of China, Hefei 230026, China. [2] CAS Key Laboratory of Microscale Magnetic Resonance, University of Science and Technology of China, Hefei 230026, China. [3] Synergetic Innovation Center of Quantum Information and Quantum Physics, University of Science and Technology of China, Hefei 230026, China. [4] School of physics, Huazhong University of Science and Technology, Wuhan 430074, China. [5] These authors contributed equally: Xi Chen, Ze Wu.
✉email: xinyoulu@hust.edu.cn; xhpeng@ustc.edu.cn

The study of superradiant phase transitions (SPT), driven by the singularity of quantum fluctuation at the critical point, has undergone tremendous developments in recent years[1]. SPT was proposed in the Dicke model, describing the collective interaction between $N$ two-level systems and a quantum field, in the thermodynamics limit $N \to \infty$[2,3]. When $N = 1$, the Dicke model is reduced to a Rabi model, in which the SPT has also been predicted theoretically by replacing the thermodynamics limit with the classical oscillator limit $\Omega/\omega \to \infty$ ($\Omega$ and $\omega$ being the frequency of spin and field, respectively)[4–6]. SPT is a special kind of quantum phase transitions, which happens along with changing the system parameters at zero temperature. When the coupling strength between the two-level system and the quantum field is increased across the quantum critical point, the ground state of the system changes abruptly, corresponding to the phase transition from a normal phase to a superradiant phase with a boost of ground-state photon number. Specifically, in the normal phase, the ground state of the cavity field is not occupied, while in the superradiant phase the ground state is macroscopically occupied and becomes twofold degenerate, corresponding to a spontaneously $\mathbb{Z}_2$ symmetry breaking. This leads to the appearance of important quantum effects in the supper-radiant phase, including spin-field entanglement, distinguishable quantum superposition with large-amplitude, and so on[7,8]. These quantum effects can play important roles in quantum metrology and quantum computation. Thus, the SPT is not only fundamentally interesting in statistical physics and electrodynamics but also has potential applications for quantum information science[9].

The cavity or circuit quantum electrodynamics (QED) systems[10,11], allowing to manipulate the light-matter interaction at the quantum level, offer an important platform of realizing Dicke model/Rabi model. However the required critical parameter regime and ultralow-temperature ground state preparation for implementing equilibrium SPT are normally hard to be satisfied with current technologies of cavity QED. More importantly, the existence of the equilibrium SPT in the cavity QED systems is still challenged by a no-go theorem[12–20]. Specifically, for describing the dipole atom-field interactions in the cavity QED system, the standard Dicke and Rabi Hamiltonians have neglected the squared term of electromagnetic vector potential (i.e., $A^2$ term), which will forbid the occurrence of equilibrium SPT. This is because the $A^2$ term, via adding a coupling-dependent potential of the cavity field, leads to the disappearance of the singularity of the quantum fluctuation in the whole parameter space. The corresponding debate on whether or not the $A^2$ term should be included in the effective models for light-matter interactions continues as of today from 1970s. Recent work put forward a SPT scheme with a hybrid circuit QED system immune to the no-go theorem[21,22] by an auxiliary squeezing term, where the Rabi Hamiltonian can be naturally realized by the interaction between a superconducting qubit and a resonator, and the auxiliary term can be introduced by the quadratic optomechanical coupling between two superconducting resonators. However, current optomechanical techniques cannot provide the strong quadratic optomechanical interaction at the single-photon level and other parameter condition for ground state SPT[23]. Considering these experimental difficulties in real cavity QED systems, quantum simulation provides an alternative and flexible technique for the experimental investigation of SPT, as well as the no-go theorem. For example, the dynamics of QRM[24–26] and nonequilibrium SPT of Dicke model have been experimentally simulated on various quantum simulation platforms, e.g., Bose-Einstein condensates[27–29] and trapped ions[30]. Very recently, the quantum phase transition of the standard QRM has been simulated with trapped ions[31]. However, the effect of the no-go theorem has not been experimentally studied so far.

In this work, by employing a nuclear magnetic resonance (NMR) quantum simulator, we simulate the effective quantum Rabi model including the $A^2$ and antisqueezing terms (approaching the classical oscillator limit $\Omega/\omega \to \infty$) by a well-defined spin-to-oscillator mapping scheme. We experimentally demonstrate the recovering of the SPT in the Rabi model with $A^2$ term by introducing the antisqueezing effect. Our work doesn't try to directly solve the long-lasting theory debate on whether or not the $A^2$ term should be included in the effective model. Based on its excellent controllability, the NMR system provides a good testbed for quantum simulations and other quantum protocols[32,33]. Interestingly, we experimentally show that the antisqueezing effect not only leads to the re-appearance of SPT when including the $A^2$ term, but also to the SPT's reversal, i.e., to the transition from normal phase (NP) to superradiant phase (SP) along with decreasing spin-field coupling strength. This originally comes from the exponentially enhanced ZPF induced by the antisqueezing effect that recovers the singularity of the ground state of the system. The optimized parameter condition including the necessary antisqueezing strength for a phase transition is identified by presenting experimentally the antisqueezing-dependent phase diagram of the ground state. In the SP, we experimentally obtain strong spin-oscillator entanglement and squeezed Schrödinger cat states of spins exhibiting a negative Wigner distribution, large-amplitude separation of peaks, and distinct interference fringes. These states could be used for fault-tolerant quantum computation[34,35] and quantum metrology[36] approaching the Heisenberg limit, aside from providing fundamental insights into the nature of decoherence and the quantum-classical transition[37]. Our work also provides the important family of antisqueezing with a new type of applications, besides widely known ones in quantum precision measurement[38] and enhancing light-matter interaction[39–46].

## Results

**Physical model.** The Rabi model with Hamiltonian

$$\hat{H}_R = \frac{\Omega}{2}\hat{\sigma}_z + \omega\hat{a}^\dagger\hat{a} + \lambda(\hat{a}^\dagger + \hat{a})\hat{\sigma}_x \qquad (1)$$

describes a two-level system with frequency $\Omega$ interacting with an oscillator mode with frequency $\omega$, and $\lambda$ denotes the coupling strength. Here $\hat{a}$ ($\hat{a}^\dagger$) is the annihilation (creation) operator of the oscillator mode, and $\hat{\sigma}_z$, $\hat{\sigma}_x$ are the Pauli operators for the two-level system. This model has the $\mathbb{Z}_2$ (or parity) symmetry associating with a well-defined parity operator $\hat{\Pi} = e^{i\pi\hat{\mathbb{N}}}$, where $\hat{\mathbb{N}} = \hat{a}^\dagger\hat{a} + (1/2)(\hat{\sigma}_z + 1)$ is the total excitation number of the system. As shown in the regime I of Fig. 1b, the ground-state SPT is predicted theoretically in the classical oscillator limit $\Omega/\omega \to \infty$, characterized by a vanishing of the lowest excitation energy[4]. However, this SPT will disappear when the $A^2$ term $\hat{H}_A = (\alpha\lambda^2/\Omega)(\hat{a} + \hat{a}^\dagger)^2$ ($\alpha \geq 1$ decided by the Thomas-Reiche-Kuhn sum rule) is included in the actual cavity QED systems (i.e., $\hat{H}_{QED} = \hat{H}_R + \hat{H}_A$), corresponding to the regime II of Fig. 1b. This is also known as the no-go theorem[12–20]. Here we introduce an antisqueezing effect into the actual cavity QED systems, i.e., considering the Rabi Hamiltonian with the $A^2$ term and an antisqueezing term $\hat{H}_{As} = -\xi(\hat{a} + \hat{a}^\dagger)^2$: $\hat{\mathbb{H}} = \hat{H}_R + \hat{H}_A + \hat{H}_{As}$. Note that formally, adding the $\hat{H}_{As}$ simply renormalizes the prefactor of the $A^2$ term. However, in the simulated system, $\hat{H}_A$ and $\hat{H}_{As}$ have a different physical meaning and a different parameter dependence on the changed parameter $\lambda$ during checking the occurrence of SPT. In the following, we will also discuss the different effects of $A^2$ term and the antisqueezing term on the SPT from the experimental results. As shown in the regime

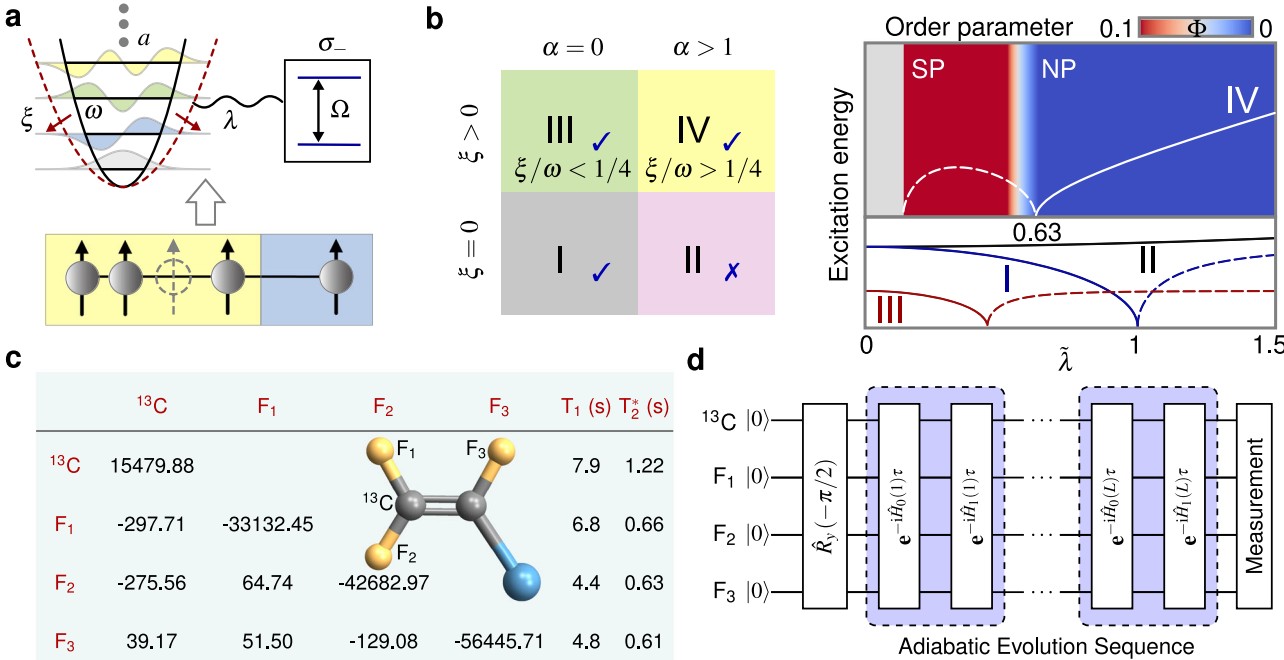

**Fig. 1 Theoretical model, phase diagram, physical system, and quantum circuit for implementing SPT. a** $N+1$ spins are used to simulate the quantum Rabi model consisting of a two-level system coupled to a Boson field with $2^N$ harmonic levels with coupling strength $\lambda$. The red arrows and dashed line-potential indicate the antisqueezing effect. **b** Phase transition property in the limit $\Omega/\omega \to \infty$, described by the lowest excitation energy and the order parameter $\Phi$ in the parameter regimes from I to IV defined by the values of $\xi$ and $\alpha$. The occurrence and disappearance of SPT are indicated by tick and cross, respectively. The solid and dashed lines correspond to the NP and SP, respectively. **c**, Molecular structure of $^{13}$C-iodotriuroethylene and the relevant parameters. The diagonal and off-diagonal elements represent chemical shifts and J-couplings (all in Hz), respectively. **d**, Quantum circuit for for implementing SPT including the adiabatic ground state preparation with $\hat{H}_0(l) = [1 - s(l)]\hat{H}_0$, $\hat{H}_1(l) = s(l)\hat{H}_s$, where $l = 0, 1, 2...L$, and $s(l)$ slowly changes from 0 to 1.

IV of Fig. 1b, the antisqueezing effect can induce the re-occurrence of SPT beyond no-go theorem. This recovering of SPT cannot be simply understood as trivially reducing the strength of $A^2$ term below a given value, i.e., leading to $\alpha < 1$ (see Supplementary Note 3). This is because the physical mechanics of generating the antisqueezing term and the $A^2$ term are different. The antisqueezing term can be realized by the quadratic opto-mechanical coupling[21,22], and it provides a modification on the potential of the bosonic field with a $\lambda$ − independent strength $\xi$. However, the $A^2$ term originally comes from the quantized light-matter interaction in the cavity QED system, whose coefficient $\alpha\lambda^2/\Omega$ is $\lambda$ − dependent and would be altered when one checks the occurrence of SPT by changing the coupling strength $\lambda$. Instead, the antisqueezing term induces the occurrence of SPT via recovering the singularity of zero-point fluctuation (ZPF) of system in the case of including $A^2$ term. The regime IV of Fig. 1b theoretically shows the re-appearance of SPT via the singularity of the excitation energy and the sudden change of the order parameter $\Phi = (\omega/\Omega)\langle\hat{a}^\dagger\hat{a}\rangle$ at the critical point $\tilde{\lambda} = \sqrt{1 + \alpha\tilde{\lambda}^2 - 4\xi/\omega}$ with $\tilde{\lambda} = 2\lambda/\sqrt{\Omega\omega}$. Specifically, the rescaled ground-state occupation of oscillator $\Phi = (1/4)(\tilde{\lambda}_s^2 - \tilde{\lambda}_s^{-2})$ becomes nonzero from $\Phi = 0$ at the critical point (see Supplementary Note 1). The regime III of Fig. 1b demonstrates that the antisqueezing effect could dramatically reduce the critical point of SPT in the case of $\alpha = 0$.

To experimentally demonstrate the ground-state SPT in the NMR platform, we simulate the Rabi model including the $A^2$ and antisqueezing terms by using $N+1$ spins, where $N$ spins simulate the oscillator and 1 spin simulates the two-level system, as shown in Fig. 1a. Based on the generators of SU(2), the mapping process

from spins to the oscillator is defined as

$$\hat{a} = \hat{A}_- \sqrt{\hat{\Sigma}_z}, \qquad \hat{a}^\dagger = \hat{A}_+ \sqrt{\hat{\Sigma}_z + \mathbb{1}^{\otimes N}}, \qquad (2)$$

where $\hat{\Sigma}_z = -\sum_{i=1}^N 2^{i-2}\hat{\sigma}_z^i + (2^N - 1)/2$, and $\mathbb{1}^{\otimes N} = \mathbb{1} \otimes \cdots \otimes \mathbb{1}$ is the identity matrix of $2^N \times 2^N$ dimensions. Here the definitions of operators $\hat{A}_\pm$ and the well-defined spin-to-oscillator mapping process are shown in Methods. This mapping process has some similarities to Holstein-Primakoff transformation, and it is exact in the limit of $N \to \infty$. We use $^{13}$C-iodotriuroethylene dissolved in d-chloroform as a 4-qubit quantum simulator, consisting of one $^{13}$C and three $^{19}$F nuclear spins, as shown in Fig. 1c. The experiments are conducted on Bruker Avance III 400 MHz spectrometer at room temperature. In the weak-coupling approximation, the natural Hamiltonian of the sample is described as

$$\hat{H}_{\text{NMR}} = \sum_{i=1}^4 \pi\nu_i\hat{\sigma}_z^{(i)} + \sum_{1\leqslant i<j\leqslant 4} \frac{\pi}{2}J_{ij}\hat{\sigma}_z^{(i)}\hat{\sigma}_z^{(j)}, \qquad (3)$$

where $\nu_i$ represents the chemical shift of the $i$-th spin, and $J_{ij}$ is the scalar coupling strength between two spins. The values of parameters $\nu_i$ and $J_{ij}$ are given in Fig. 1c. The system, initially at the thermal equilibrium state, is first prepared to a pseudo-pure state (PPS) $\hat{\rho}_{\text{pps}} = [(1 - \varepsilon)/16]\mathbb{1}^{\otimes 4} + \varepsilon|0\rangle\langle 0|$ by using the selective-transition approach[47]. Here $\varepsilon \approx 10^{-5}$ is the polarization, and the experimental fidelity of $\hat{\rho}_{\text{pps}}$ is about 0.991. The detailed initialization process is shown in Supplementary Note 5.

**Antisqueezing-enhanced ZPF.** Let us first experimentally demonstrate the antisqueezing-enhanced ZPF in our 4-spin system, which is the key physical mechanism of recovering ground-state SPT

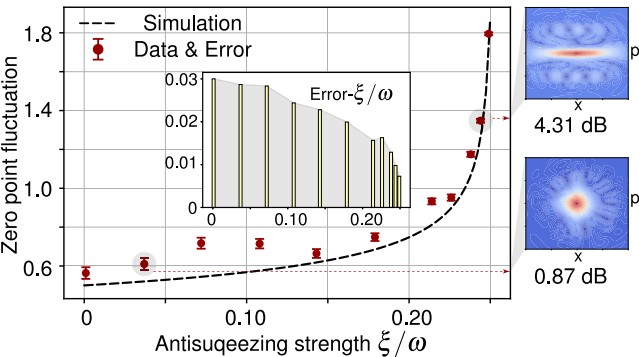

**Fig. 2 Experimental measured ZPF (red circles) versus the antisqueezing strength $\xi$, demonstrating exponentially enhanced ZPF of the oscillator by the antisqueezing effect.** The theoretical expectation is shown in the dashed line. The inset plot shows the standard deviation of ZPF for each data point. The measured Wigner functions of two squeezed vacuum states are shown in the right side.

in the case of including the $A^2$ term. It is also the nature of antisqueezing-enhanced light-matter interaction explored in recent theoretical [39–45] and experimental [46] works. Theoretically, the antisqueezing term $\hat{H}_{As}$ will make the ground state of an oscillator from a vacuum state $|0\rangle$ to the squeezed vacuum state $\hat{S}(r)|0\rangle$ with $\hat{S}(r) = \exp[r(\hat{a}^2 - \hat{a}^{\dagger 2})/2]$ and the squeezing parameter $r = (1/4)\ln(1 - 4\xi/\omega)$. Then the ZPF, defined as ZPF = $\sqrt{\langle \hat{x}^2 \rangle - \langle \hat{x} \rangle^2}$ with dimensionless quadrature $\hat{x} = (\hat{a} + \hat{a}^\dagger)/2$, will be exponentially enhanced with increasing the antisqueezing strength $\xi$ (see the dashed line of Fig. 2). In our experiment, the truncated squeezing operator in the Hilbert space $\{|n\rangle\}$ is implemented with a gradient ascent pulse engineering (GRAPE) pulse with duration $15ms$[48]. Then, we perform a three-step measurement to obtain the ZPF of the squeezing vacuum state. (i) The expected values $\langle \hat{a}^\dagger \hat{a} + \hat{a}\hat{a}^\dagger \rangle$ are obtained by measuring the diagonal elements of the squeezing vacuum state, i.e., separately applying a $\pi/2$ readout pulse $\hat{R}_y^{(i)}(\pi/2) = \exp(-i\pi/4 \hat{\sigma}_y^{(i)})$ to each of the four qubits after a pulsed field gradient and then measuring the resulting NMR spectrum of the corresponding qubit $i$. (ii) The expected values $\langle \hat{a} + \hat{a}^\dagger \rangle$ are obtained by measuring the fourth qubit with/without a readout operator $\hat{U}_1 = \sum_{n=0}^{14}|n\rangle\langle n + 1| + |15\rangle\langle 0|$. (iii) The expected values $\langle \hat{a}^2 + \hat{a}^{\dagger 2} \rangle$ are obtained by measuring the third qubit with/without a readout operator $\hat{U}_2 = \sum_{n=0}^{13}|n\rangle\langle n + 2| + |14\rangle\langle 0| + |15\rangle\langle 1|$. Here the operators $\hat{U}_1$ and $\hat{U}_2$ are well-designed to transfer all the measured elements to the observables that can be read out directly by the NMR signals, which are also realized by GRAPE pulses. Due to the used sample in natural abundance, i.e., only 1% of the molecules had a $^{13}C$ nuclear spin, we read out all four spins via the $^{13}C$ channel, by applying SWAP gates and measuring the $^{13}C$ spin. The experimental data shown in Fig. 2 are in good agreement with the theoretical predictions. From the inset of Fig. 2, one can also find that the error bars, coming from the statistical fluctuation of the NMR spectra, become smaller along with increasing the antisqueezing effect. This originates from the amplitude enhancement of ZPF with large antisqueezing strength, which can be useful in quantum metrology. To clearly show the antisqueezing effect, we also present the Wigner functions of two experimentally reconstructed states by quantum state tomography[49].

**Recovering of SPT in the case of including $A^2$ term.** Next we shall experimentally simulate the equilibrium SPT beyond no-go

theorem induced by the antisqueezing effect. With the exact squeezing transformation, the ground state of the total system Hamiltonian $\hat{\mathbb{H}} = \hat{H}_R + \hat{H}_A + \hat{H}_{As}$ is equivalent to apply a squeezing operation $\hat{S}(\tilde{r})$ on the ground state of the transformed Hamiltonian $\hat{\mathbb{H}}_s = \hat{S}^\dagger(\tilde{r})\hat{\mathbb{H}}\hat{S}(\tilde{r}) = (\Omega/2)\hat{\sigma}_z + \omega_s\hat{a}^\dagger\hat{a} + \lambda_s(\hat{a}^\dagger + \hat{a})\hat{\sigma}_x$. Here the different effect of $A^2$ term and antisqueezing term on the SPT have been transferred into the interconnected coefficients $\omega_s = e^{2\tilde{r}}\omega$, $\lambda_s = e^{-\tilde{r}}\lambda$, $\tilde{r} = (1/4)\ln(1 + \alpha\tilde{\lambda}^2 - 4\xi/\omega)$, and the constant term in Hamiltonian $\hat{\mathbb{H}}_s$ has been ignored. Now the problem is transferred to experimentally preparing the ground state of $\hat{\mathbb{H}}_s$. In our sample, $^{13}C$ spin is labeled as the two-level system, and three $^{19}F$ nuclear spins are used to map the truncated boson mode $\hat{a}$ with the defined mapping process in Eq. (2). In the experiments, we employ the widely used adiabatic method[50] to prepare the ground state of $\hat{\mathbb{H}}_s$. According to the quantum circuit in Fig. 1d, the 4-spin sample is firstly prepared into the ground state of Hamiltonian $\hat{\mathbb{H}}_0 = \sum_{i=1}^{4}\hat{\sigma}_y^{(i)}$ by applying $\pi/2$ pulses along $y$ axis on PPS for four spins simultaneously. Then the quantum system is controlled to adiabatically evolve under the instantaneous Hamiltonian $\hat{\mathbb{H}}(l) = [1 - s(l)]\hat{\mathbb{H}}_0 + s(l)\hat{\mathbb{H}}_s$, with $l = 1, 2, \ldots L$ and $s(l)$ changing slowly from 0 to 1. The system will finally evolve to the ground state of $\hat{\mathbb{H}}_s$, denoted by $|G\rangle_s$, after the above adiabatic evolution. The experimental adiabatic evolution is implemented by the GRAPE pulse with duration 26 ms. Based on the prepared ground state $|G\rangle_s$, the order parameter of SPT, expressed as $\Phi = (\omega/\Omega)\left(\cosh(2\tilde{r})\langle\hat{a}^\dagger\hat{a}\rangle_s - (1/2)\sinh(2\tilde{r})(\langle\hat{a}^{\dagger 2}\rangle_s + \langle\hat{a}^2\rangle_s) + \sinh^2\tilde{r}\right)$, can be obtained by measuring the corresponding expectations defined with $|G\rangle_s$. The detailed measurement process is shown in Methods.

To show the occurrence of SPT, we present the dependence of the order parameter $\Phi$ on $\tilde{\lambda}$ for different frequency ratio $\Omega/\omega$ in Fig. 3a, b, along with the theoretical expectations. Figure 3a shows that, without antisqueezing effect ($\xi = 0$), the phase transition is forbidden by the $A^2$ term, i.e., the no-go theorem. However, the SPT is recovered by introducing a fixed antisqueezing effect ($\xi/\omega = 0.26$) in Fig. 3b. Specifically, the experimental order parameter $\Phi$ in Fig. 3b changes from almost zero to finite number at $\tilde{\lambda} \approx 0.63$ with decreasing $\tilde{\lambda}$, which indicates a reversed SPT. This reversed SPT originally comes from the nontrivial competition between the introduced antisqueezing term and the $A^2$ term, and it is quite different from the case of simulating normal SPT in the Rabi model with an $A^2$ term at different intensities (see Supplementary Note 3). Along with increasing $\Omega/\omega$, the dependence of $\Phi$ on $\tilde{\lambda}$ approaches the case of $\Omega/\omega \rightarrow \infty$, where the reversed SPT occurs exactly at the critical point $\tilde{\lambda} = \sqrt{1 + \alpha\tilde{\lambda}^2 - 4\xi/\omega}$ (see the inset of Fig. 3b). Notice that the available Hilbert space for simulating the oscillator is limited in the experiments, which makes the values of the order parameter in the SP smaller with growing values $\Omega/\omega$. But the experimental results are enough to demonstrate the rapid growth of the order parameter near the critical point. Physically, the occurrence of SPT in our experiment originally comes from the recovering of the singularity of ground-state fluctuations due to the antisqueezing-enhanced ZPF of the system, as shown in Supplementary Fig. 2.

Actually, we had also performed the corresponding simulation experiment without postprocessing by physically implementing the squeezing operation $\hat{S}(\tilde{r})$ in different parameter regions, and obtained the order parameter $\Phi$ by experimentally preparing the

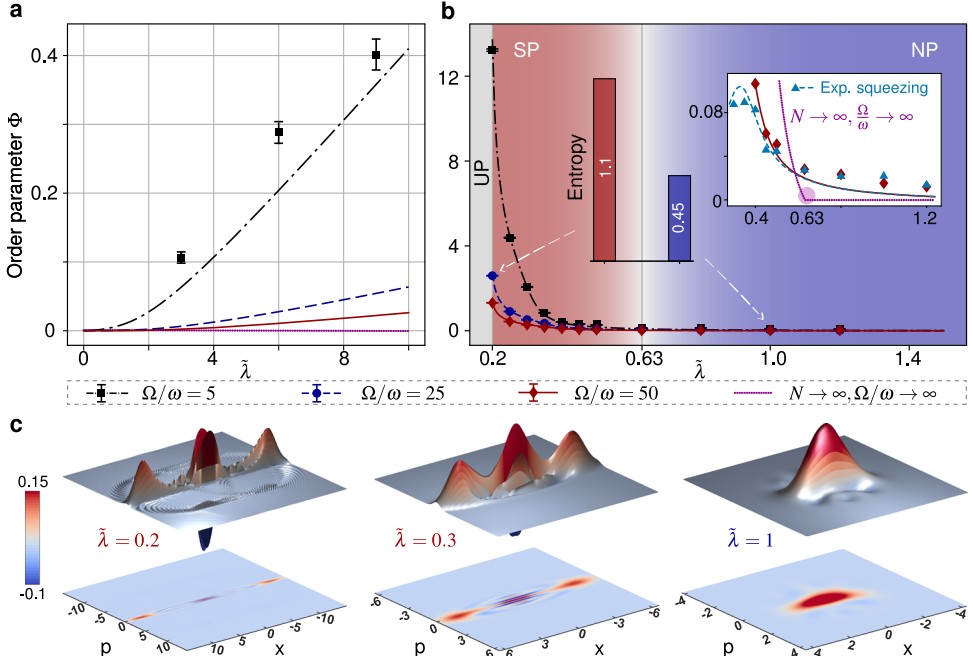

**Fig. 3 Experimental demonstration of the recovering of equilibrium SPT induced by the antisqueezing effect. a, b** The order parameter $\Phi$ versus the scaled spin-field coupling strength $\tilde{\lambda}$ for **a** $\xi = 0$ and for **b** $\xi/\omega = 0.26$. The inset plot of **b** shows the comparison between the cases of finite-parameter ($\Omega/\omega = 50$) and the classical oscillator limit ($\Omega/\omega \to \infty$, the SPT occurs exactly). The red solid line (blue dotted line) represents the order parameter $\Phi$ on $\tilde{\lambda}$ without (with) the experimental implementation of the squeezing operation. The shading of **b** divides normal phase (NP) and superradiant phase (SP) according to the exact critical point $\tilde{\lambda} = \sqrt{1 + \alpha\tilde{\lambda}^2 - 4\xi/\omega}$. The system is in unstable phase (UP) when the rescaled ground-state excitation becomes an imaginary number, i.e., $\tilde{\lambda} < \sqrt{(1/\alpha)(4\xi/\omega - 1)}$. The bar graphs of **b** indicate two von Neumann entropies of system in SP ($\tilde{\lambda} = 0.2$) and NP ($\tilde{\lambda} = 1$) when $\Omega/\omega = 25$. **c,** The corresponding Wigner functions of experimentally prepared ground states for $\tilde{\lambda} = 0.2,\ 0.3$ (SP) and $\tilde{\lambda} = 1$(NP) when $\Omega/\omega = 25$. Here $\alpha = 1.1$ for all figures.

ground state of the original Hamiltonian $\hat{\mathbb{H}}$. The corresponding experimental results are also plotted in the inset of Fig. 3b as well as in Supplementary Figure 3a. It is clearly seen that there is a similar rapid growth of the order parameter $\Phi$ near the critical point, indicating the occurrence of SPT, for both the cases with and without postprocessing. The order parameter $\Phi$ without postprocessing in deep superradiant phase will drop down because the truncated squeezing operator in small-size Hilbert space with large squeezing parameter $r$ will loss the validity (see Supplementary Figure 4). The method of the postprocessing avoids this limit of the restricted Hilbert space in realizing $\hat{S}(\tilde{r})$.

**Entangled and squeezed Schrödinger cat states in superradiant phase.** Rich quantum resources can be obtained in the superradiant phase, such as the quantum entanglement and quantum superposition of coherent states, i.e., Schrödinger cat states. They are significant for quantum metrology and quantum computation, aside from their fundamental nature. In the limit $\Omega/\omega \to \infty$, the ground state of our system (including the antisqueezing term) is theoretically predicted as a squeezed state $|G\rangle_{\text{np}} = \hat{S}(\tilde{r}_{\text{np}})|0\rangle_a|\downarrow\rangle$ in the NP and a spin-oscillator entangled state $|G\rangle_{\text{sp}} \approx (1/\sqrt{2})\hat{S}(\tilde{r})[\hat{D}(\beta)|0\rangle_a|\downarrow\rangle_+ + \hat{D}(-\beta)|0\rangle_a|\downarrow\rangle_-]$ in the SP with a defined displaced operator $\hat{D}(\beta)$ (see Supplementary Note 1). By quantum state tomography, we experimentally reconstruct the ground states of the system, when it is in the NP ($\tilde{\lambda} = 1$) and SP ($\tilde{\lambda} = 0.2, 0.3$). The bar graphs in Fig. 3b clearly demonstrate that the strong entanglement is obtained in the SP

via the von Neumann entropy $S = -\text{tr}(\hat{\rho}_a\log_2\hat{\rho}_a)$ ($\hat{\rho}_a$ is the reduced density matrix of oscillator). Moreover, in the SP, the entangled state $|G\rangle_{\text{sp}}$ becomes a squeezed cat state when we measure the $^{13}$C spin in the $(1/\sqrt{2})(|\downarrow\rangle_+ \pm |\downarrow\rangle_-)$ basis (see Supplementary Note 2). We plot the corresponding Wigner functions of three experimentally reconstructed ground states in Fig. 3c, which clearly show the appearance of squeezed cat states in the SP. They have a negative Wigner distribution with distinct interference fringes and large size for $\tilde{\lambda} = 0.3$ and 0.2, which is a key factor for implementing super-resolution metrology with high probability and fault-tolerant quantum computing. Note that, the Schrödinger cat states also can be experimentally prepared via the homodyne detection on the fock states[51,52], photon subtraction on the squeezed state[53,54], and high-order nonlinear atom-field interaction[55] and so on. Here the realization of quantum superposition of spin coherent states indicates a spontaneously $\mathbb{Z}_2$ breaking, which is evidenced by the nonzero ground-state coherence $\langle\hat{a}\rangle_{\text{sp}}$.

To fully demonstrate the rich equilibrium dynamics induced by the antisqueezing effect, we present the experimental ground-state phase diagram characterized by the rescaled ground-state excitation $\Phi$ in Fig. 4a, b. The realization of reversed SPT are shown again, and it also can be seen from the reduced density matrix $\hat{\rho}_a^s$ reconstructed experimentally by quantum state tomography. As shown in Fig. 4c, the main contributions to $\Phi$, i.e., $\langle\hat{a}^\dagger\hat{a}\rangle_s$ (diagonal elements) and $\langle\hat{a}^2\rangle_s$ (sub-sub diagonal elements), approximately change from zero to finite value along with decreasing $\tilde{\lambda}$. Figure 4a again shows that the dependence of

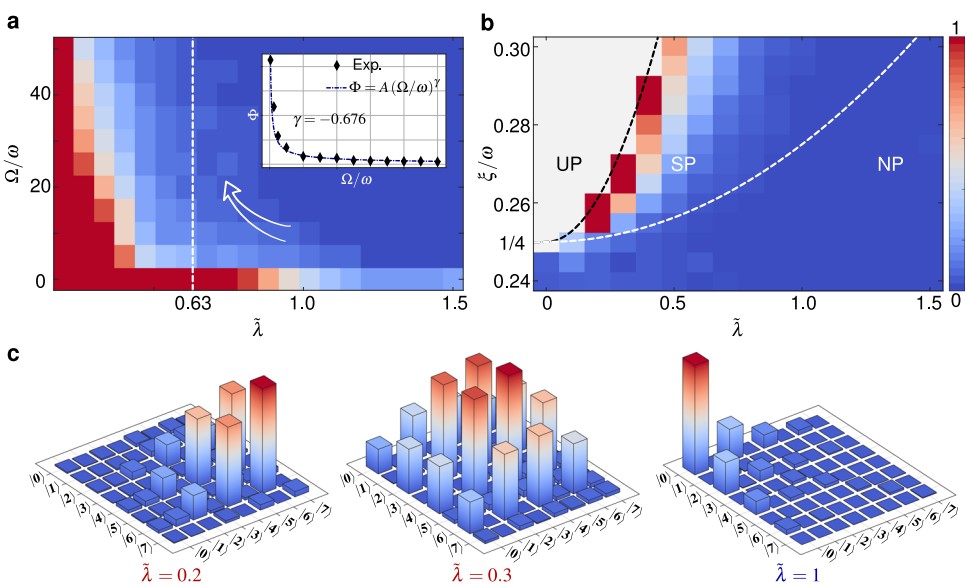

**Fig. 4 Modulated ground-state phase diagram by the antisqueezing. a** The measured order parameter $\Phi$ versus $\tilde{\lambda}$ and $\Omega/\omega$ for $\xi/\omega = 0.26$. The inset plot shows the experimental power-law scaling of $\Phi$ at the critical point, and the fitted finite-frequency scaling exponent is $\gamma = -0.676$. The white dashed line corresponds to the exact critical point in the limit $\Omega/\omega \to \infty$. **b** The dependence of $\Phi$ on $\tilde{\lambda}$ and $\xi/\omega$ for $\Omega/\omega = 20$. The white (black) dashed line corresponds to the critical parameter distinguishing NP and SP (SP and UP). **c**, The reduced density matrix $\hat{\rho}_a^s = \text{tr}_\sigma(|G\rangle_s \langle G|)$ for $\Omega/\omega = 25$, obtained by the experimentally reconstructed ground state of $\hat{\mathbb{H}}_s$. The diagonal elements contribute to $\langle \hat{a}^\dagger \hat{a} \rangle_s$, while the sub-sub diagonal elements contribute to $\langle \hat{a}^{\dagger 2} \rangle_s$ and $\langle \hat{a}^2 \rangle_s$. Other parameters are the same as that in Fig. 3.

$\Phi$ on $\tilde{\lambda}$ approaches to the case of exactly occurring SPT at the quantum critical point along with increasing $\Omega/\omega$. Furthermore, we measure a series of order parameters $\Phi$ at the critical point for different values of $\Omega/\omega$, showing the finite-frequency scaling for the observable $\Phi$ in the inset of Fig. 4a. The order parameter $\Phi$ vanishes with a power-law scaling, and the fitted finite-frequency scaling exponent $\gamma = -0.676$ is very close to the universal exponent $-2/3$ of the Rabi and Dicke model, which verifies the experimental realization of SPT in finite-frequency regime again. By fixing the value of $\Omega/\omega$, Fig. 4b indicates that $\xi/\omega > 1/4$ is required for the occurrence of phase transition, which is consistent with the analytical parameter condition $\tilde{\lambda} \geqslant \sqrt{1 + \alpha \tilde{\lambda}^2 - 4\xi/\omega}$ of SPT. However, too large antisqueezing strength will bring the system into an unstable phase (UP), when the rescaled ground-state excitation becomes an imaginary number, i.e., $1 + \alpha \tilde{\lambda}^2 - 4\xi/\omega < 0$, corresponding to $\tilde{\lambda} < \sqrt{(1/\alpha)(4\xi/\omega - 1)}$. In other words, the coupling $\tilde{\lambda}$ cannot be reduced arbitrarily as the Hamiltonian would become unbounded from below. With increasing the spin-oscillator coupling $\tilde{\lambda}$, the competition between the $A^2$ and the antisqueezing effect will push the system into SP when $\tilde{\lambda} > \sqrt{1 + \alpha \tilde{\lambda}^2 - 4\xi/\omega}$, and then into NP when $\tilde{\lambda} < \sqrt{1 + \alpha \tilde{\lambda}^2 - 4\xi/\omega}$. Our experimental results are basically consistent with the theoretical boundaries of different phases. This consistency becomes better and better along with increasing $\Omega/\omega$ (see Supplementary Fig. 1), which again predicts the occurring of SPT in the classical oscillator limit.

## Discussion

In summary, from a physical point of view, we have presented a first proof-of-principle experiment to demonstrate equilibrium SPT beyond no-go theorem induced by the antisqueezing effect. Our experiment is not in contradiction with the original no-go theorem, and it indeed relaxes the limit of $A^2$ term on SPT via introducing the antisqueezing effects. To understand the re-

appearance of SPT, we experimentally show the enhanced ZPF by antisqueezing, which ultimately recovers the singularity of the ground state of the system. The modulated ground-state phase diagram by the antisqueezing is experimentally obtained by preparing the ground state of the system with the adiabatic method and measuring the order parameter by a hybrid method of combining the experimental data and a theoretical priori. Associating with the SPT, we also experimentally realize the strong entanglement and the squeezed cat state of spins, which provides new possibilities for both quantum metrology and quantum information processing. Our work is fundamentally interesting in demonstrating that the $A^2$ term is not the ultimate limit for experimental observation of equilibrium SPT in the cavity QED system. The current scheme is suitable not only for NMR systems but will also works well in other physical systems, such as trapped ions [56] and NV centers [57]. They open new routes for experimentally exploring the novel quantum optical effect with the platform of NMR or other spin systems.

## Methods

**Spin-to-oscillator mapping scheme.** Generally, $N$ qubits can be used to simulate a boson mode with $2^N$ levels by arranging all spin states as the binary form of the corresponding excitation number:

$$
\begin{aligned}
|0\rangle &\mapsto |\uparrow_N \uparrow_{N-1} \cdots \uparrow_1 \uparrow_0\rangle = |00 \cdots 00\rangle, \\
|1\rangle &\mapsto |\uparrow_N \uparrow_{N-1} \cdots \downarrow_1 \uparrow_0\rangle = |00 \cdots 01\rangle, \\
&\vdots \quad \vdots \\
|2^N - 1\rangle &\mapsto |\downarrow_N \downarrow_{N-1} \cdots \downarrow_1 \downarrow_0\rangle = |11 \cdots 11\rangle.
\end{aligned}
\tag{4}
$$

This scheme makes sure the spin space is fully utilized and the spin matrices are exactly the same as the mapped oscillator operators. The mathematical form of this mapping scheme has some similarities to Holstein-Primakoff transformation. We will first give the mapping representation of the truncated number operator

$$
\hat{a}^\dagger \hat{a} = -\sum_{i=1}^N 2^{i-2} \hat{\sigma}_z^{(i)} + \frac{2^N - 1}{2},
\tag{5}
$$

where the superscript $(i)$ denotes the $i$-th qubit. Eq. (5) can be proved by the mathematical induction.

**Proof**. Proof of Eq. (5). Obviously, the equation establishes when $N = 1$. Assume Eq. (5) is true for $N = k$. Now for $N = k + 1$,

$$\text{diag}\{0, \cdots, 2^{k+1} - 1\}$$
$$= \text{diag}\{0, \cdots, 2^k - 1\} \otimes \mathbb{1} + 2^k \mathbb{1}^{\otimes k} \otimes \left(\frac{\mathbb{1} - \hat{\sigma}_z^{(k+1)}}{2}\right)$$
$$= \left(\frac{2^k - 1}{2} - \sum_{i=1}^{k} 2^{i-2} \hat{\sigma}_z^{(i)}\right) \otimes \mathbb{1} \qquad (6)$$
$$- 2^{k-1}\left(\mathbb{1}^{\otimes k} \otimes \hat{\sigma}_z^{(k+1)} - \mathbb{1}^{\otimes (k+1)}\right)$$
$$= -\sum_{i=1}^{k+1} 2^{i-2} \hat{\sigma}_z^{(i)} + \frac{2^{k+1} - 1}{2},$$

where $\mathbb{1}^{\otimes k} = \mathbb{1} \otimes \cdots \otimes \mathbb{1}$ is the identity matrix of $2^k \times 2^k$ dimensions. Then the formula will be true for every natural number $N$.

To obtain the representations of operators $\hat{a}$ and $\hat{a}^\dagger$, let's define $\hat{\Sigma}_z \equiv \hat{a}^\dagger \hat{a}$ with $\hat{\Sigma}_z|n\rangle = n|n\rangle$, and the 'increasing operator' ('decreasing operator') $\hat{A}_+ / (\hat{A}_-)$ as follows

$$\hat{A}_+ \equiv \hat{\sigma}_+^{(1)} + \hat{\sigma}_-^{(2)}\hat{\sigma}_+^{(1)} + \cdots + \hat{\sigma}_+^{(N)}\hat{\sigma}_-^{(N-1)} \cdots \hat{\sigma}_-^{(1)},$$
$$\hat{A}_- \equiv \hat{\sigma}_-^{(1)} + \hat{\sigma}_+^{(1)}\hat{\sigma}_-^{(2)} + \cdots + \hat{\sigma}_+^{(1)}\hat{\sigma}_+^{(2)} \cdots \hat{\sigma}_-^{(N)}. \qquad (7)$$

It is not difficult to find that $\hat{A}_+|n\rangle = |n+1\rangle$ and $\hat{A}_-|n+1\rangle = |n\rangle$ for all $0 \leq n < 2^N - 1$. The above definitions allow us to construct the truncated creation and annihilation operators conveniently. Based on these properties, we have

$$\hat{A}_- \sqrt{\hat{\Sigma}_z}|n\rangle = \sqrt{n}|n-1\rangle,$$
$$\hat{A}_+ \sqrt{\hat{\Sigma}_z + \mathbb{1}^{\otimes N}}|n\rangle = \sqrt{n+1}|n+1\rangle, \qquad (8)$$

which leads to

$$\hat{a} = \hat{A}_- \sqrt{\hat{\Sigma}_z},$$
$$\hat{a}^\dagger = \hat{A}_+ \sqrt{\hat{\Sigma}_z + \mathbb{1}^{\otimes N}}. \qquad (9)$$

Thus Eq. (5) and Eq. (9) form the spin-to-oscillator mapping scheme in our quantum simulation experiments. It is worth noting that the mapping scheme involves multi-body interactions which is not easy to be simulated. By a combination of the natural NMR Hamiltonian and radio frequency control pulses, we implement the mapping scheme in NMR systems[58,59].

**Order parameter measurement**. In order to demonstrate the occurrence of the SPT in an effective way with the limited qubits, we adopt an alternative experimental scheme by means of the postprocessing (see Supplementary Note 3). Since we have $\hat{\mathbb{H}}_s = \hat{S}^\dagger(\bar{r})\hat{\mathbb{H}}\hat{S}(\bar{r})$, the ground states of $\hat{\mathbb{H}}$ and $\hat{\mathbb{H}}_s$ are linked by a squeezing transformation

$$|G\rangle = \hat{S}(\bar{r})|G\rangle_s, \qquad (10)$$

where $|G\rangle$ and $|G\rangle_s$ are the ground states of $\hat{\mathbb{H}}$ and $\hat{\mathbb{H}}_s$, respectively. Then the order parameter of SPT can be expressed as

$$\Phi = (\omega/\Omega)\langle G|\hat{a}^\dagger \hat{a}|G\rangle$$
$$= (\omega/\Omega)_s\langle G|\hat{S}^\dagger(\bar{r})\hat{a}^\dagger \hat{a}\hat{S}(\bar{r})|G\rangle_s \qquad (11)$$

Together with the following derivation

$$\hat{S}^\dagger(\bar{r})\hat{a}^\dagger \hat{a}\hat{S}(\bar{r}) \equiv \hat{S}^\dagger(\bar{r})\hat{a}^\dagger \hat{S}(\bar{r})\hat{S}^\dagger(\bar{r})\hat{a}\hat{S}(\bar{r})$$
$$= \cosh(2\bar{r})\hat{a}^\dagger \hat{a} - \frac{1}{2}\sinh(2\bar{r})(\hat{a}^{\dagger 2} + \hat{a}^2) + \sinh^2\bar{r}, \qquad (12)$$

we obtain

$$\Phi = (\omega/\Omega)\cosh(2\bar{r})\langle \hat{a}^\dagger \hat{a}\rangle_s$$
$$- (1/2)\sinh(2\bar{r})\left(\langle \hat{a}^{\dagger 2}\rangle_s + \langle \hat{a}^2\rangle_s\right) + \sinh^2\bar{r}. \qquad (13)$$

This means the order parameter of SPT can be obtained by measuring the corresponding expectations in the ground state $|G\rangle_s$.

In short, according to Eq. (5) with $N = 3$, the value of $\langle a^\dagger a\rangle_s$, corresponding to $\langle \hat{\sigma}_z^{(i)}\rangle_s$, can be obtained by measuring the diagonal elements of the ground state $|G\rangle_s$. Similarly, the boson operators $\hat{a}^{\dagger 2} + \hat{a}^2$ can be expressed as

$$\hat{a}^{\dagger 2} + \hat{a}^2 = \sum_{l,m \in \{0,1\}} \sqrt{c_{lm}(c_{lm} - 1)}|l\rangle\langle l| \otimes \hat{\sigma}_x^{(2)} \otimes |m\rangle\langle m|$$
$$+ \sqrt{3 \times 4}(|100\rangle\langle 010| + |010\rangle\langle 100|) \qquad (14)$$
$$+ \sqrt{4 \times 5}(|101\rangle\langle 011| + |011\rangle\langle 101|).$$

The expectation of the first term is read out directly from the second qubit in Eq. (14). For the last two terms, i.e., $|100\rangle\langle 010| + |010\rangle\langle 100|$ and

$|101\rangle\langle 011| + |011\rangle\langle 101|$, we can first transfer them into the observable of the second qubit (namely $\hat{\sigma}_x^{(2)}$) by applying a designed operation $\hat{U}_2$ to the system. Subsequently, the corresponding expectations are obtained by measuring the second qubit again. Here the operation $\hat{U}_2$ can be implemented by the quantum circuit shown in Supplementary Figure 5 or an equivalent GRAPE pulse.

## Data availability
The data that support the findings of this study are available from the corresponding authors upon reasonable request.

## Code availability
The codes for numerical simulation and data processing are available from the corresponding authors upon reasonable request.

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

## Acknowledgements

This work is supported by the National Key R & D Program of China (Grants No. 2018YFA0306600 and 2016YFA0301203), the National Science Foundation of China (Grants No. 11822502, 11974125, and 11927811), Anhui Initiative in Quantum Information Technologies (Grant No. AHY050000).

## Author contributions

X.P. and X.L. conceived the project. X.L. conceived the relevant theoretical constructs. X.P., X.C., and Z.W. designed the experiment. X.C. and Z.W. performed the measurements and analyzed the data. M.J. assisted with the experiment. X.P. and J.D. supervised the experiment. X.L. wrote the draft. All authors contributed to analyzing the data, discussing the results, and writing the manuscript.

## Competing interests

The authors declare no competing interests.
