## [Peer Review File · Nature Communications]

Reviewers' Comments:

Reviewer #1:

Remarks to the Author:

This paper presents an experimental implementation of the quantum Rabi model using nuclear spins. The essential idea is to encode the lowest states of a bosonic mode into the combined Hilbertspace of multiple nuclear spins. Using this scheme, the authors show how optimized RF pulses can be applied to implement effective squeezing terms and Rabi-type interactions. With these control capabilities, an initial spin state is adiabatically transferred into the ground state of the Rabi-model, which includes the so-called A2 term and an additional anti-squeezing term. Using this technique, clear signatures of the superradiant phase transition and characteristic features, such as entangled ground states are observed.

This is a very interesting paper, which demonstrates in an impressive way how nuclear spins can be used to simulate the physics of spin-boson systems. The presented results are convincing and highlight various non-trivial features of the superradiant phase transition. I think this is an important experiment for the whole field of quantum simulation and well-suited for a publication in Nature Communications.

However, before publication I ask the author to change the way how their findings are presented. The long lasting debate about the equilibrium superradiant phase transition and the no-go-theorem is not about whether or not the A2 term prevents this transition. It is about whether or not this term should be included in effective models for light-matter interactions. Since the current experiment is a simulation where all terms in the Hamiltonian are added by hand, such an experiment cannot provide any insights on this matter. Also, although there are no losses in this system, it is not in equilibrium. The nuclear spins are prepared in the ground state of an effective Hamiltonian, which is engineered by applying time-dependent driving fields. In essence, this is the same experimental situation as in other quantum simulations schemes with cold atoms or trapped ions (see, e.g, Ref. [17]). Given the long and controversial history of this topic one should clearly distinguish between the ground state of an effective (driven) model and a true equilibrium situation, where the A2 but no squeezing terms are present.

Reviewer #2:

Remarks to the Author:

I have read with interest the manuscript by Xi Chen et al. entitled "Experimental quantum simulation of superradiant phase transition beyond no-go theorem via antisqueezing", where an NMR implementation of the ground state (GS) of the quantum Rabi model (QRM) is reported. The QRM, describing the interaction between a two-level system and a bosonic field, is known to exhibit a quantum phase transition for a critical value of the atom-field coupling, in the limit where the ratio of the frequencies of the two-level system and the field approaches infinity. The authors experimentally explore these critical phenomena using an NMR system consisting of molecules containing 4 nuclear spin-1/2s. To that end, they map a truncated QRM Hamiltonian to the Hilbert space spanned by the 4 nuclear spins. This Hamiltonian is adiabatically introduced while the system is initially in its GS, such that the system transitions to the GS of the QRM. By measuring second-order moments of the field operators they can reconstruct an order parameter consisting of a rescaled phonon occupation number and study its behavior for different values of the coupling strength including its critical value, as well as different values of the ratio of frequencies. The authors also offer a full-reconstruction of the density matrix of the field as well as its Wigner function at different parameter regimes. Additionally, they demonstrate their ability to perform a squeezing operation on the simulated field degrees of freedom. All of this aims at evidencing the presence of the quantum phase transition.

The manuscript is extensive and detailed, and the methodology and derivations are technically sound. In that sense, I believe that the provided data is complete and of good quality, meeting the requirements for reproducibility. However, I find that the framing of the problem and the interpretation of the results are inadequate and that this significantly reduces their relevance. Additionally, the clarity of the presentation is suboptimal, in a big part due to a rather poor quality

of the writing that hinders the communication of the results, and requires a big effort from the reader to sense the authors' meaning.

My main concern is regarding the framing of the experiment as breaking a no-go theorem. On page 2, the authors state that

"However, this SPT will disappear when the A^2 term $H_A = (\alpha\lambda^2/\Omega)(a+a^\dagger)^2$ ($\alpha \geq 1$ decided by the Thomas-Reiche-Kuhn sum rule) is included in the actual cavity QED systems, corresponding to the regime II of FIG. 1b. This is the no-go theorem of SPT, and the corresponding debate continues to today from 1970s. Here we will demonstrate experimentally the above no-go theorem can be broken through by introducing an antisqueezing effect, i.e., $H_{AS} = -\xi(a+a^\dagger)^2$ in the platform of NMR."

which comes to say that the no-go theorem is broken if the term proportional to $(a+a^\dagger)^2$ is reduced below a given value. This seems to be a trivial statement as the theorem already considers that the phase transition disappears only under the assumption that the A^2 term is strong enough. Thus, I would say that this does not constitute a breaking of the theorem as it is.

Besides, the authors introduce both the A^2 term and the antisqueezing term by transforming the QRM Hamiltonian under a squeezing operator. Indeed the authors are only able to introduce the sum of both terms, and thus, the distinction between the two appears to be artificially inserted to motivate the discussion on the interplay between the no-go theorem and the antisqueezing. In my view, what the authors are simulating is the QRM with an A^2 term at different intensities, and showing that the phase transition disappears when this term exceeds a particular value. In that sense, the authors are confirming the no-go theorem rather than going beyond it.

Apart from that, the experiment is a nice implementation of the GS of the QRM at different parameter regimes. However, this is not the first time that the QRM has been simulated, for example

[1] Braumüller, J. et al. Analog quantum simulation of the rabi model in the ultra-strong coupling regime. Nat. Commun. 8, 779 (2017).

[2] Langford, N. K. et al. Experimentally simulating the dynamics of quantum light and matter at deep-strong coupling. Nat. Commun. 8, 1715 (2017)

[3] Lv, D. et al. Quantum simulation of the quantum rabi model in a trapped ion. Phys. Rev. X 8, 021027 (2018).

nor the first time that the quantum phase transition has been explored, see

[4] Cai, ML., Liu, ZD., Zhao, WD. et al. Observation of a quantum phase transition in the quantum Rabi model with a single trapped ion. Nat Commun 12, 1126 (2021).

Thus, the novelty of the reported result relies on the platform in which it has been achieved. By this, I don't imply that it is of less value, as the techniques required to implement the model in a different system may be of value in themselves. However, as a simulator of the QRM and its phase transition, the NMR setup does not seem to bring any significant advantage over previously simulated setups. On the opposite, the fact that the available Hilbert space is smaller makes the results less convincing. For example, in Fig. 3b and Fig. 4a, the values of the order parameter in the SP become smaller with growing values of the ratio between the frequency of the two-level system and the field. That is somehow surprising to me, as I would expect the phase transition to become sharper. I guess this is explained precisely by the small size of the considered Hilbert space, but it somehow challenges the validity of these figures of merit as indicative of the phase transition. Perhaps the authors could comment on that.

Additionally, I have a few other comments:

- It seems to me that the experiment consists of two parts. In the first part, the authors show the

implementation of the squeezing operation on a Hilbert space spanned by 4 qubits. In the next part, they simulate the QRM on the same 4 qubits, but now the bosonic field spans only 3 qubits, as one needs to be left for the two-level system in the QRM. However, these two experiments seem to be unrelated. It is implied that since the QRMs with and without the A^2 term are related by a squeezing transformation, the simulation of the latter from the former requires the implementation of the squeezing operation. However, if I understand correctly, in the experiment, this is never done. The authors compute the expectation values of the GS of the QRM with the A^2 term from the expectation values of the QRM without the A^2 term. In that sense, I would say that the phase transition is only implemented for the QRM without the A^2 term, and the analysis of the effect of the A^2 term is introduced via post-processing. In this regard, I find it surprising that the authors never present results for the simpler case of the QRM without the A^2 term, that is without post-processing. Is there any reason for that?

- In Fig. 3b, I would argue that the bar graph indicating the values of the entropy in the two different phases is a bit misleading, as these are placed next to each other around the critical value and seem to indicate that the entropy changes its value abruptly, while they correspond to quite distant values of the coupling strength.

- The Wigner functions of the reconstructed GS look quite smooth for a Hilbert space of dimension 8. Are these fitted curves? How are these obtained?

In summary, for the reasons stated above, I do not believe that the paper merits publication in Nature Communications.

Reviewer #3:

Remarks to the Author:

The authors report the implementation of an experiment designed to show the occurrence of equilibrium Superradiant phase transition (SPT) beyond the no-go theorem using the NMR technique. The nuclear spins of a small molecule were used to simulate a two-level system coupled to a truncated Harmonic oscillator according to the Hamiltonian H given in Eq 1. According to ref. 10, when an additional term is introduced in the Hamiltonian SPT disappears (this is the no-go theorem). In the present paper, it was shown theoretically (in the supplementary material) and by an "experimental simulation" that the no-go theorem is broken by introducing an antisqueezing effect, i.e. introducing a new term in the Hamiltonian.

The experiment is not a demonstration of the SPT but a quantum simulation of the effect. Here the dynamic of the system is mimicked by numerically optimized shaped r.f. pulses. The authors stated in the text that the experimental conditional to observe SPT is hard to achieve in QED. In this context, the implementation of a quantum simulation makes sense, but I think that this should be set clear from the beginning, adding a comment to the abstract.

The experiment was implemented using standard NMR techniques and seems to be well done. More details should be added about the data analyses. How the Wigner functions were reconstructed from the density matrices? Usually, only the deviation of the density matrix is obtained in NMR tomography and it is not suitable to reconstruct the Wigner function. The authors probably have used some postprocessing procedure to reach the density matrices in figure 4 and Wigner functions in 3. I think that the authors should introduce, at least in the supplementary material, more about data analyses.

Concerning the relevance of the paper, I have two points that should be addressed. First, the relevance of SPT should be better explained. What is SPT? Why is it important? A better introduction to SPT would improve the text for nonspecialist readers and highlight the relevance of the paper. Another important question is: Could the present scheme be implemented in QED systems? How the Hamiltonian that is implemented here by a grape pulse would be implemented in a real QED system?

The second point is the claim that the present scheme provides a general method to enhance NMR signal. The present experiment is a simulation of a specific Hamiltonian, using r.f. pulses that were optimized for a specific small molecule. It is interesting that the antisqueezing term "works transferring polarization" from the quantum field to the spin but I do not see how this could be used in a general NMR experiment. Furthermore, the signal enhancement in Fig S5 seems to be around a factor of 4. We should reach the same factor using a simple experiment to transfer polarization from Fluorine to Carbon.

For the points raised above I think that the paper is not suitable for Nature Communications in the present form.

A minor comment:

The coupling constants in figure 1c do not have the units.

List of main changes

Main text:

1. We have added section and subsection titles.
2. In the introduction (Paragraphs 1-3), we have added more background of SPT in quantum Rabi/Dicke model and no-go theorem, including the basic description, the importance of SPT and the experimental progress in the quantum simulation of quantum Rabi model.
3. In the second paragraph, we have added some comments on how to implement our scheme in a real QED system.
4. In the paragraph above Eq. (2), we emphasize the nontrivial physical mechanism of our work and the essential difference between the A^2 term and the antisqueezing term.
5. We have modified the inset of Fig. 2 and the corresponding description about SNR enhancement in the abstract and in the main text (see the end of the first subsection titled '*Antisqueezing-enhanced ZPF*' in Results).

Supplementary material:

1. In Sec. II, we have added more details about how to calculate and plot the Wigner functions.
2. We have performed two new experiments and numerical simulations, which are displayed in the newly added section (Sec. III of the supplemental material) for the detailed comparison between the SPT with and without the implementation of the squeezing operation, as well as the comparison between our work and the case of QRM without the A^2 term.
3. In Sec. III and IV, we have added a discussion of the validity of truncated squeezing operation used in our experiment.
4. In Sec. VI, we have added more details about the data analyses, including the quantum state tomography and measurement procedure.
5. In Sec. VII, we have added more detailed explanation about the ZPF enhancement induced by the anti-squeezing effect.

We have also carefully polished the language of the manuscript and all changes are marked with color highlighting.

Reply to Reviewer #1

Comment 1-1

This paper presents an experimental implementation of the quantum Rabi model using nuclear spins. The essential idea is to encode the lowest states of a bosonic mode into the combined Hilbert space of multiple nuclear spins. Using this scheme, the authors show how optimized RF pulses can be applied to implement effective squeezing terms and Rabi-type interactions. With these control capabilities, an initial spin state is adiabatically transferred into the ground state of the Rabi-model, which includes the so-called A^2 term and an additional anti-squeezing term. Using this technique, clear signatures of the superradiant phase transition and characteristic features, such as entangled ground states are observed.

This is a very interesting paper, which demonstrates in an impressive way how nuclear spins can be used to simulate the physics of spin-boson systems. The presented results are convincing and highlight various non-trivial features of the superradiant phase transition. I think this is an important experiment for the whole field of quantum simulation and well-suited for a publication in Nature Communications.

Our response: We sincerely thank the reviewer for the favorable recommendation for our work.

Comment 1-2

However, before publication I ask the author to change the way how their findings are presented. The long lasting debate about the equilibrium superradiant phase transition and the no-go-theorem is not about whether or not the A^2 term prevents this transition. It is about whether or not this term should be included in effective models for light-matter interactions. Since the current experiment is a simulation where all terms in the Hamiltonian are added by hand, such an experiment cannot provide any insights on this matter.

Our response: As the reviewer points out, the long lasting debate about the equilibrium superradiant phase transition (SPT) and the no-go-theorem is about whether or not the A^2 term should be included in the effective models for light-matter interactions. The original description of the no-go theorem is that the equilibrium SPT will not happen if the A^2 term is included in the effective models for light-matter interactions (Ref. [14]). Different from the previous theoretical works, trying to settle this dispute directly, here our experiment shows the physical phenomenon of the recovery of the SPT by introducing the antisqueezing term in QRM when the A^2 term exists in the real systems. It is indeed different from the direct solving of the long lasting theory debate on the no-go theorem, i.e., whether or not the A^2 term should be included in the effective model.

According to the reviewer's suggestion, we have changed the presenting way of our findings in this revised manuscript. See the second paragraph of the main text:

".....The corresponding debate that whether or not the A^2 term should be included in the effective models for light-matter interactions continues to today from 1970s....."

And also the third paragraph of the main text:

".....Our work doesn't try to directly solve the long lasting theory debate on whether or not the A^2 term should be included in the effective model....."

Comment 1-3

Also, although there are no losses in this system, it is not in equilibrium. The nuclear spins are prepared in the ground state of an effective Hamiltonian, which is engineered by applying time-dependent driving fields. In essence, this is the same experimental situation as in other quantum simulations schemes with cold atoms or trapped ions (see, e.g, Ref. [17]). Given the long and controversial history of this topic one should clearly distinguish between the ground state of an effective (driven) model and a true equilibrium situation, where the A^2 but no squeezing terms are present.

Our response: We agree that we use the ground state of an effective NMR (driven) model to simulate the equilibrium SPT. By using the quantum simulator, we clearly demonstrate the antisqueezing effect induced SPT in the case of including A^2 term.

According to the reviewer's suggestion, we have added the corresponding statement in this revised manuscript (see the third paragraph of main text):

"Due to these experimental difficulties in real cavity QED systems, quantum simulation thus provides an alternative and flexible technique for the experimental investigation of SPT as well as the no-go theorem.....In this article, by employing nuclear magnetic resonance (NMR) quantum simulator, we simulate the effective quantum Rabi model including the A^2 and antisqueezing terms....."

Reply to Reviewer #2

Comment 2-1

I have read with interest the manuscript by Xi Chen et al. entitled "Experimental quantum simulation of superradiant phase transition beyond no-go theorem via antisqueezing", where an NMR implementation of the ground state (GS) of the quantum Rabi model (QRM) is reported. The QRM, describing the interaction between a two-level system and a bosonic field, is known to exhibit a quantum phase transition for a critical value of the atom-field coupling, in the limit where the ratio of the frequencies of the two-level system and the field approaches infinity. The authors experimentally explore these critical phenomena using an NMR system consisting of molecules containing 4 nuclear spin-1/2s. To that end, they map a truncated QRM Hamiltonian to the Hilbert space spanned by the 4 nuclear spins. This Hamiltonian is adiabatically introduced while the system is initially in its GS, such that the system transitions to the GS of the QRM. By measuring second-order moments of the field operators they can reconstruct an order parameter consisting of a rescaled phonon occupation number and study its behavior for different values of the coupling strength including its critical value, as well as different values of the ratio of frequencies. The authors also offer a full-reconstruction of the density matrix of the field as well as its Wigner function at different parameter regimes. Additionally, they demonstrate their ability to perform a squeezing operation on the simulated field degrees of freedom. All of this aims at evidencing the presence of the quantum phase transition.

The manuscript is extensive and detailed, and the methodology and derivations are technically sound. In that sense, I believe that the provided data is complete and of good quality, meeting the requirements for reproducibility. However, I find that the framing of the problem and the interpretation of the results are inadequate and that this significantly reduces their relevance. Additionally, the clarity of the presentation is suboptimal, in a big part due to a rather poor quality of the writing that hinders the communication of the results, and requires a big effort from the reader to sense the authors' meaning.

Our response: We thank the reviewer's patiently reading and summary on our work, and pointed out the advantages and weakness of our manuscript. We have tried to our best to improve the presentation of the writing including the framing of the problem and the interpretation of the results to increase its readability and avoid potential ambiguities.

Comment 2-2

My main concern is regarding the framing of the experiment as breaking a no-go theorem. On page 2, the authors state that "However, this SPT will disappear when the A^2 term $\hat{H}_A = (\alpha\lambda^2/\Omega)(\hat{a} + \hat{a}^\dagger)^2$ ($\alpha \geq 1$ decided by the Thomas-Reiche-Kuhn sum rule) is included in the actual cavity QED systems, corresponding to the regime II of Fig. 1b. This is the no-go theorem of SPT, and the corresponding debate continues to today from 1970s. Here we will demonstrate experimentally the above no-go theorem can be broken through by introducing an antisqueezing effect, i.e., $\hat{H}_{As} = -\xi(\hat{a} + \hat{a}^\dagger)^2$ in the platform of NMR." which comes to say that the no-go theorem is broken if the term proportional to $(\hat{a} + \hat{a}^\dagger)^2$ is reduced below a given value. This seems to be a trivial statement as the theorem already considers that the phase transition disappears only under the assumption that the A^2 term is strong enough. Thus, I would say that this does not constitute a breaking of the theorem as it is.

Our response: We think this referee misunderstood the nontrivial physical mechanism in the simulated physical model, i.e., how does the antisqueezing term recover SPT in the case of including A^2 term $\hat{H}_A = (\alpha\lambda^2/\Omega)(\hat{a} + \hat{a}^\dagger)^2$ with $\alpha > 1$. In the physical model simulated in the experiment, the antisqueezing term we introduced here doesn't trivially reduce the value of α in the A^2 term. From the view of mathematics, the antisqueezing term plays a role of making α become effectively λ -dependent, i.e., $\hat{H}'_A = \alpha'(\lambda)\frac{\lambda^2}{\Omega}(\hat{a} + \hat{a}^\dagger)^2$ with $\alpha'(\lambda) = \alpha - \xi\Omega/\lambda^2$.

From the view of physics, this is nontrivial. On the one hand, the physical mechanism of generating the A^2 term and the antisqueezing term in the simulated physical model are different. The A^2 term originally comes from the quantized light-matter interaction in the cavity QED system, whose coefficient $\alpha\lambda^2/\Omega$ ($\alpha \geq 1$ decided by the Thomas-Reiche-Kuhn sum rule [14]) is λ -dependent. However the antisqueezing term can be realized by the quadratic optomechanical coupling [21,22], and it provides a modification on the potential of the bosonic field with a λ -independent strength ξ . The antisqueezing term induces the occurrence of SPT via recovering the singularity of zero-point fluctuation (ZPF) of system in the case of including A^2 term. The original definition of no-go theorem describes that the SPT disappears when A^2 with $\alpha > 1$ is included [14]. Our quantum simulation experiment is going to demonstrate the interesting physical phenomenon, i.e., the SPT is recovered in the case of including A^2 term with $\alpha > 1$ by introducing an antisqueezing term. In this sense, we state that our work simulated the SPT beyond no-go theorem induced by antisqueezing.

Figure 1: **a**, Blue line: reversed SPT with A^2 term and antisqueezing term; Red line: Normal SPT by reducing α to below 1. **b**, Normal SPT simulated in the Rabi model with A^2 term at different intensities.

On the other hand, the effect of the λ -dependent coefficient $\alpha'(\lambda)$ on the SPT is different from that of a fixed α coefficient without the antisqueezing term. This is because one should check the occurrence of SPT in the total parameter range of λ . If the introduced antisqueezing term trivially reduces α below 1 to recover the SPT in the case of including A^2 term, then we should obtain the normal phase transition, as denoted by red line in Fig. 1a of this reply. However, the *reversed* SPT is expected, i.e., the phase transition to the superradiant phase happens with the decreasing of the spin-field coupling strength, as denoted by the blue line in Fig. 1a. This paradox comes from the misunderstanding of the role of antisqueezing term. The derived critical point $\tilde{\lambda} = \sqrt{1 + \alpha\tilde{\lambda}^2 - 4\xi/\omega}$ (i.e., $\tilde{\lambda} = \sqrt{\frac{4\xi/\omega - 1}{\alpha - 1}}$) in our work also clearly shows that the role of antisqueezing term cannot be understood simply as reducing the value of α , i.e., the strength of A^2 term. Actually, in our work, the antisqueezing effect recovers the SPT in the case of including A^2 term with $\alpha > 1$ by increasing the zero-point fluctuation (ZPF) of system and recovering its singularity, as shown in Fig. 2 of main text and Fig. S2a of the supplementary material.

To make this point clearer, we have added the corresponding discussion (see the paragraph above Eq. (2) of main text):

".....This recovering of SPT cannot be simply understood as trivially reducing the strength of A^2 term below a given value, i.e., leading to $\alpha < 1$. This is because the physical mechanics of generating the antisqueezing term and the A^2 term are different....."

We also did some additional experiments to demonstrate the essential difference between our model and the normal SPT (see the new Sec. III in the supplementary material).

Comment 2-3

Besides, the authors introduce both the A^2 term and the antisqueezing term by transforming the QRM Hamiltonian under a squeezing operator. Indeed the authors are only able to introduce the sum of both terms, and thus, the distinction between the two appears to be artificially inserted to motivate the discussion on the interplay between the no-go theorem and the antisqueezing. In my view, what the authors are simulating is the QRM with an A^2 term at different intensities, and showing that the phase transition disappears when this term exceeds a particular value. In that sense, the authors are confirming the no-go theorem rather than going beyond it.

Our response: We do not agree with the opinion that "the authors are confirming the no-go theorem rather than going beyond it." As stated in the response of Comment 2-2, for the physical model simulated in our experiment, the introduced antisqueezing term doesn't trivially reduce the value of α in the A^2 term. Firstly, the physical mechanism of generating the A^2 term and the antisqueezing term in the simulated physical model are different. Moreover, the antisqueezing term makes α become effectively λ -dependent, whose effect on the SPT is different from that of a fixed α coefficient without the antisqueezing term. By quantum simulation of the physical model with the A^2 term and the antisqueezing term, our experiment indeed shows the occurrence of SPT beyond no-go theorem by introducing the antisqueezing effect. Moreover, the reversed SPT obtained in our experiments is quite different from the Rabi model with A^2 term at different intensities, i.e., the normal SPT shown in Fig. 1b of this reply. In this sense, we state that our work simulated the SPT beyond no-go theorem induced by antisqueezing.

To make this point clearer, we have added the corresponding discussion (see the paragraph above Eq. (2) of main text):

".....This recovering of SPT cannot be simply understood as trivially reducing the strength of A^2 term below a given value, i.e., leading to $\alpha < 1$. This is because the physical mechanics of generating the antisqueezing term and the A^2 term are different....."

Also see the revised paragraph under Fig. 4:

".....This reversed SPT originally comes from the nontrivial competition between the introduced antisqueezing term and A^2 term, and it is quite different from the case of simulating normal SPT in the Rabi model with an A^2 term at different intensities."

Comment 2-4

Apart from that, the experiment is a nice implementation of the GS of the QRM at different parameter regimes. However, this is not the first time that the QRM has been simulated, for example

[1] Braumüller, J. et al. Analog quantum simulation of the rabi model in the ultra-strong coupling regime. Nat. Commun. 8, 779 (2017).

[2] Langford, N. K. et al. Experimentally simulating the dynamics of quantum light and matter at deep-strong coupling. Nat. Commun. 8, 1715 (2017)

[3] Lv, D. et al. Quantum simulation of the quantum rabi model in a trapped ion. Phys. Rev. X 8, 021027 (2018).

nor the first time that the quantum phase transition has been explored, see

[4] Cai, ML., Liu, ZD., Zhao, WD. et al. Observation of a quantum phase transition in the quantum Rabi model with a single trapped ion. Nat Commun 12, 1126 (2021).

Thus, the novelty of the reported result relies on the platform in which it has been achieved. By this, I don't imply that it is of less value, as the techniques required to implement the model in a different system may be of value in themselves. However, as a simulator of the QRM and its phase transition, the NMR setup does not seem to bring any significant advantage over previously simulated setups.

Our response: We really thank that the reviewer pointed out these related references, which have been cited in the revised manuscript. Refs. [1,2,3] simulated the dynamics of QRM, while Ref. [4] studied the quantum phase transition of the standard QRM. We agree that our experiment is not the first one that the QRM has been simulated, but the first one to simulate the occurrence of equilibrium SPT beyond no-go theorem which is not just the implementation of the GS of the QRM at different parameter regimes, as shown in the responses of **Comment 2-2** and **Comment 2-3**. Concretely, we simulate QRM including the A^2 term and antisqueezing term, and observe the occurrence of SPT beyond no-go theorem (i.e., in the case of including A^2 term) by introducing the antisqueezing term for the first time. The nontrivial competition between the A^2 term and the introduced antisqueezing term ultimately leads to some new and interesting results, which are not reported in Ref. [4]. For example, we experimentally observe the *reversed* SPT, i.e., the phase transition to the superradiant phase happens as decreasing the spin-field coupling strength. The novelty of our results relies on both experimental quantum simulation of the new physical model and physical phenomenon, rather than the platform in which it has been achieved. In this sense, as commented by Reviewer #1, "this is an important experiment for the whole field of quantum simulation". Besides, we would like to mention that Ref. [4] (arXiv:2102.05409 submitted on 10 Feb 2021) and our work (arXiv:2102.07055 submitted on 14 Feb 2021) are posted in the arXiv almost at the same time.

In order to better show the novelty of our work, we have modified the introduction to discuss the status of experimental quantum simulations for QRM (see the third paragraph of main text):

".....Very recently, quantum phase transition of the standard QRM has been simulated on trapped ions. However, the no-go theorem has not been experimentally studied so far. In this article, by employing nuclear magnetic resonance (NMR) quantum simulator, we simulate the effective quantum Rabi model including the A^2 and antisqueezing terms (approaching the classical oscillator limit $\Omega/\omega \rightarrow \infty$) by a well-defined spin-to-oscillator mapping scheme....."

We have also added corresponding detailed discussion in the newly added Sec. III of the supplementary material.

Comment 2-5

On the opposite, the fact that the available Hilbert space is smaller makes the results less convincing. For example, in Fig. 3b and Fig. 4a, the values of the order parameter in the SP become smaller with growing values of the ratio between the frequency of the two-level system and the field. That is somehow surprising to me, as I would expect the phase transition to become sharper. I guess this is explained precisely by the small size of the considered Hilbert space, but it somehow challenges the validity of these figures of merit as indicative of the phase transition. Perhaps the authors could comment on that.

Our response: The available Hilbert space used in the NMR platform is indeed limited, which makes the values of the order parameter in the SP become smaller with growing values Ω/ω . But the Hilbert space used in our experiment is enough to demonstrate the main results of our work, which will not make our results less convincing.

The main result of our experiment is to demonstrate the re-occurrence of SPT induced by the antisqueezing effect. Theoretically, the phase transition is decided by the sudden change of the order parameter *at the critical point*, i.e., the order parameter suddenly boosts from zero at the critical point. In our experiment, all experimental data in Fig. 3b of the main text show the obvious growth of the order parameter near the critical point, which clearly demonstrates the occurrence of SPT in the finite parameter regime. The asymptotic behavior of SPT in terms of Ω/ω can be reached by employing the larger Hilbert space for simulating the Boson field.

To avoid the ambiguity, we added the relevant discussion in the revised manuscript (see the modified paragraph under Fig. 4 of main text):

"Notice that the available Hilbert space used in the NMR simulator is limited, which makes the values of the order parameter in the SP become smaller with growing values Ω/ω . But the experiment results are enough to demonstrate the sudden change of the order parameter at the critical point."

Comment 2-6

Additionally, I have a few other comments:

- It seems to me that the experiment consists of two parts. In the first part, the authors show the implementation of the squeezing operation on a Hilbert space spanned by 4 qubits. In the next part, they simulate the QRM on the same 4 qubits, but now the bosonic field spans only 3 qubits, as one needs to be left for the two-level system in the QRM. However, these two experiments seem to be unrelated. It is implied that since the QRMs with and without the A^2 term are related by a squeezing transformation, the simulation of the latter from the former requires the implementation of the squeezing operation. However, if I understand correctly, in the experiment, this is never done. The authors compute the expectation values of the GS of the QRM with the A^2 term from the expectation values of the QRM without the A^2 term. In that sense, I would say that the phase transition is only implemented for the QRM without the A^2 term, and the analysis of the effect of the A^2 term is introduced via post-processing. In this regard, I find it surprising that the authors never present results for the simpler case of the QRM without the A^2 term, that is without post-processing. Is there any reason for that?

Our response: First of all, the two parts of the experiments are related to each other. The main purpose of the first part is to experimentally demonstrate the exponentially enhanced ZPF of the oscillator by the antisqueezing effect, which is the key physical mechanism that leads to the

re-occurrence of SPT in the case of including A^2 term. The second part is the main result of our work, i.e., the occurrence of SPT beyond no-go theorem via the antisqueezing effect. In order to better show the enhancement of ZPF by the antisqueezing effect, we fully utilize the qubit resource of the current quantum register in the experiment.

Actually, we had done the implementation of the squeezing operation in the experiments, and the experimental data had been shown in the supplemental material (see Fig. S3b of the submitted version). Moreover, we have performed a new experiment for the different value of Ω/ω and the new experimental data have been added in Fig. S3a in the supplemental material of current version. Since the truncated squeezing operator in small Hilbert space with large squeezing parameter r will loss the validity (see Fig. S4), the order parameter in deep superradiant phase drops down. To illustrate the effect of the truncated squeezing operator more clearly, we numerically simulate the behaviors of the SPT with the truncated squeezing operator of different-size Hilbert space, showing the disappearance of the decreasing behavior in deep superradiant phase when the large enough Hilbert space is used (see Fig. S3c).

In order to demonstrate the occurrence of the SPT in an effective way with the limited qubits, we adopt an alternative experimental scheme by means of the post-processing. The final experimental results, i.e., the order parameter of \hat{H} is reconstructed from two parts of the expectation values under the ground state of \hat{H}_s , i.e., $\langle a^\dagger a \rangle_s$ and $\langle a^2 + a^{\dagger 2} \rangle_s$, as shown in Eq. (13) of the main text. Now we have plotted the raw data of $\langle a^\dagger a \rangle_s$ and $\langle a^2 + a^{\dagger 2} \rangle_s$ in Fig. S3b. These data without post-processing is not just the order parameter of QRM without the A^2 term, and the SPT of \hat{H} can never be obtained with only the experimental results of QRM without A^2 term, i.e., $\langle a^\dagger a \rangle_s$. It needs to clarify that the scheme via the post-processing is not just equivalent to trivially simulate QRM without A^2 term. To further show their difference, we also added another new experiment for the QRM without the A^2 term. The experimental result is shown in Fig. S3d, which is obviously different from our experimental results (Fig. S3a) or even the data before post-processing (Fig. S3b).

Inspired by the review's suggestions, we have added the related description in Methods (see the second paragraph of Methods):

"In order to demonstrate the occurrence of the SPT in an effective way with the limited qubits, we adopt an alternative experimental scheme by means of the post-processing."

We also performed the new experiments and numerical simulations in the new section (Sec. III of the supplemental material) for the detailed comparison between the SPT with and without the implementation of the squeezing operation, as well as the comparison between our work and the case of QRM without the A^2 term.

Comment 2-7

- In Fig. 3b, I would argue that the bar graph indicating the values of the entropy in the two different phases is a bit misleading, as these are placed next to each other around the critical value and seem to indicate that the entropy changes its value abruptly, while they correspond to quite distant values of the coupling strength.

Our response: We thank the reviewer for his/her carefully reading and have modified the figure in this revised version.

Comment 2-8

- The Wigner functions of the reconstructed GS look quite smooth for a Hilbert space of dimension 8. Are these fitted curves? How are these obtained?

Our response: The calculation of the Wigner functions is based on the Wigner transform of any matrix element $|n\rangle\langle m|$, which is represented as

$$W_{mn}(x, p) = \sqrt{\frac{m!}{n!}} e^{i(m-n) \arctan(p/x)} \frac{(-1)^m}{\pi \hbar} \left(\frac{x^2 + p^2}{\hbar/2} \right)^{(n-m)/2} L_m^{n-m} \left(\frac{x^2 + p^2}{\hbar/2} \right) e^{-(x^2 + p^2)/\hbar},$$

with a Laguerre polynomial L_m . Then the Wigner function of any quantum state like $\rho = \sum_{n,m} r_{n,m} |n\rangle\langle m|$ can be easily achieved as $W(x, p) = \sum_{n,m} r_{n,m} W_{mn}(x, p)$. Therefore, the smoothness of the Wigner functions is closely related with the distributions of x and p values we choose to calculate, which actually has nothing to do with the dimension of quantum states. Accordingly, from the experimentally reconstructed ground states of system, one can the Wigner functions. In fact, we plot the Wigner functions by QuTiP package of Python, which is a very convenient tool. Although the space of Boson operators a and a^\dagger are of infinite dimension, the numerical tools can calculate Wigner functions from the matrix representation of states in the truncated Fock basis. In QuTiP, the truncated Boson operators is simply got by the codes ‘qt.create(N)’ and ‘qt.destroy(N)’, and the Wigner functions of states of finite dimension can be plotted by “qt.wigner(rho, xvec, pvec)”.

Here is some pivotal codes:

```
1 import numpy as np
2 import qutip as qt
3 from scipy.io import loadmat
4 import matplotlib.pyplot as plt
5
6 # Load the experimental data of quantum tomography.
7 data = loadmat('***.mat')
8 rho = qt.Qobj(data['***'])
9
10 # Calculate the Wigner function, 'xvec' and 'pvec' are the points we choose
    to calculate.
11 xvec = np.linspace(-15, 15, 300)
12 pvec = np.linspace(-15, 15, 300)
13 W = qt.wigner(rho, xvec, pvec)
14
15 # Plot the Wigner function.
16 fig = plt.figure()
17 ax = plt.axes(projection='3d')
18 X, Y = np.meshgrid(xvec, pvec)
19 ax.plot_surface(X, Y, W, rstride = 1, cstride = 1, alpha=0.8, cmap='coolwarm')
20 ax.contourf(X, Y, W, 200, zdir='z', offset=-0.24, cmap='coolwarm')
21 plt.show()
```

We have added the details about how we draw these Wigner functions in Sec. II of the supplementary material.

Comment 2-9

In summary, for the reasons stated above, I do not believe that the paper merits publication in Nature Communications.

Our response: As shown our response above, here we would like to emphasize the novelty and importance of our work again: the significant progress on experimental quantum simulation of QRM as well as the nontrivial physical mechanism introduced in experiment (see the Cover letter to Editor). Therefore, we are confident that this revised manuscript now merits publication in Nature Communications.

Reply to Reviewer #3

Comment 3-1

The authors report the implementation of an experiment designed to show the occurrence of equilibrium Superradiant phase transition (SPT) beyond the no-go theorem using the NMR technique. The nuclear spins of a small molecule were used to simulate a two-level system coupled to a truncated Harmonic oscillator according to the Hamiltonian H given in Eq 1. According to ref. 10, when an additional term is introduced in the Hamiltonian SPT disappears (this is the no-go theorem). In the present paper, it was shown theoretically (in the supplementary material) and by an "experimental simulation" that the no-go theorem is broken by introducing an antisqueezing effect, i.e. introducing a new term in the Hamiltonian.

Our response: We thank the reviewer for her/his pertinent comments about our work.

Comment 3-2

The experiment is not a demonstration of the SPT but a quantum simulation of the effect. Here the dynamic of the system is mimicked by numerically optimized shaped r.f. pulses. The authors stated in the text that the experimental conditional to observe SPT is hard to achieve in QED. In this context, the implementation of a quantum simulation makes sense, but I think that this should be set clear from the beginning, adding a comment to the abstract.

Our response: According to the review's suggestion, we have added the corresponding statements into the abstract of this revised manuscript:

"On the other hand, the experimental condition to study equilibrium SPT as well as no-go theorem is hard to achieve with current accessible technology in cavity QED system. Based on the platform of nuclear magnetic resonance (NMR) quantum simulator, here we experimentally simulate the occurrence of equilibrium SPT beyond no-go theorem by introducing the antisqueezing effect."

Also in the second paragraph:

"However the required critical parameter regime and ultralow-temperature ground state preparation for implementing equilibrium SPT are normally hard to be satisfied with current technologies of cavity QED."

Comment 3-3

The experiment was implemented using standard NMR techniques and seems to be well done. More details should be added about the data analyses. How the Wigner functions were reconstructed from the density matrices? Usually, only the deviation of the density matrix is obtained in NMR tomography and it is not suitable to reconstruct the Wigner function. The authors probably have used some postprocessing procedure to reach the density matrices in figure 4 and Wigner functions in 3. I think that the authors should introduce, at least in the supplementary material, more about data analyses.

Our response: As this referee understood, we first obtained only the deviation of the density matrix $\rho_{\Delta} = \rho - 1/2^n$ in NMR tomography, which can't be regarded as a quantum state. Therefore, we used the post-processing procedure to reach the density matrices in Fig. 4 by introducing the constraints of the normalization $\text{Tr}(\rho) = 1$, the hermiticity $\rho = \rho^{\dagger}$ and the

positive semi-definiteness $\rho \geq 0$. All of these constraints were realized by CVX toolbox in Matlab. Therefore, the reconstructed matrices will satisfy all the requirements of the density matrix of a quantum state. And the Wigner functions in Fig. 3 were obtained from the reduced density matrix of the bosonic field by tracing out the two-level system, i.e., a 8×8 density matrix. To calculate and plot the Wigner functions, we have used the QuTiP package in Python, which is a very convenient tool for numerical analysis. See the response of Comment 2-8 for the more details.

According to the review's suggestion, in the revised manuscript, we have added the corresponding descriptions regarding the data analysis to reconstruct the density matrices of quantum states in Sec. VI of supplementary material and to plot the Wigner functions in Sec. II of supplementary material:

"In the experiment, we also perform full state tomography to reconstruct the density matrices of the squeezed vacuum states. In NMR setup, we first obtained only the deviation of the density matrix $\rho_{\Delta} = \rho - \mathbb{1}/2^n$ in NMR tomography, which can't be regarded as a quantum state. Therefore, we used the post-processing procedure to reach the density matrices in Fig. 4....."

".....By employing the QuTiP package in Python, we could calculate the Wigner functions from the matrix representation of states in the truncated Fock basis, which are experimentally reconstructed by state tomography."

Comment 3-4

Concerning the relevance of the paper, I have two points that should be addressed. First, the relevance of SPT should be better explained. What is SPT? Why is it important? A better introduction to SPT would improve the text for nonspecialist readers and highlight the relevance of the paper.

Our response: We thank the reviewer for the helpful suggestions to improve this manuscript. According to the review's suggestions, we have added the description about the background of SPT and no-go theorem, including the definition, the importance and the related experimental progress in the first paragraph of the revised main text:

"SPT is a special kind of quantum phase transitions which happens along with changing the system parameters at zero temperature. When the coupling strength between the two-level system and the quantum field is increased across the quantum critical point, the ground state of the system changes abruptly, corresponding to the phase transition from a normal phase to a superradiant phase with a boost of ground-state photon number....."

As well as:

"These quantum effect are the key resources for quantum metrology and quantum computation. Thus, the SPT is not only fundamentally interesting in statistical physics and electrodynamics, but also has potential applications for quantum information science."

Comment 3-5

Another important question is: Could the present scheme be implemented in QED systems? How the Hamiltonian that is implemented here by a grape pulse would be implemented in a real QED system?

Our response: In principle, the present scheme could be implemented in a hybrid circuit QED system, where the quadratic optomechanical coupling between two superconducting resonators could introduce the antisqueezing effects, together with the Rabi Hamiltonian between a superconducting qubit and one resonator. However, current optomechanical techniques cannot provide the strong quadratic optomechanical interaction at the single photon level and other parameter condition for ground state SPT.

In order to clearly illustrate this point, we have added some related description in the second paragraph of the revised manuscript:

"Recent work put forward a SPT scheme with hybrid circuit QED system immune to no-go theorem by an auxiliary squeezing term, where the Rabi Hamiltonian can be naturally realized by the interaction between a superconducting qubit and a resonator, and the auxiliary term can be introduced by the quadratic optomechanical coupling between two superconducting resonators. However, current optomechanical techniques cannot provide the strong quadratic optomechanical interaction at the single photon level and other parameter condition for ground state SPT."

Comment 3-6

The second point is the claim that the present scheme provides a general method to enhance NMR signal. The present experiment is a simulation of a specific Hamiltonian, using r.f. pulses that were optimized for a specific small molecule. It is interesting that the antisqueezing term "works transferring polarization" from the quantum field to the spin but I do not see how this could be used in a general NMR experiment. Furthermore, the signal enhancement in Fig. S5 seems to be around a factor of 4. We should reach the same factor using a simple experiment to transfer polarization from Fluorine to Carbon.

Our response: We are very sorry to cause the misunderstanding due to our unclear statement. In fact, the antisqueezing term "does not work transferring polarization" from the quantum field to the spin, while it only "works transferring polarization" within the quantum field. It is completely different from polarization transfer experiment from Fluorine to Carbon. Here we explain how the antisqueezing effect can enhance measurement signal in our experiment. When the squeezing transformation $\hat{S}(r)$ is applied on the squeezed vacuum state, one gets

$$\hat{S}(r)|0\rangle = \frac{1}{\sqrt{\cosh r}} \sum_{n=0}^{\infty} (-\tanh r)^n \frac{\sqrt{(2n)!}}{2^n n!} |2n\rangle.$$

By means of the spin-to-oscillator mapping scheme (see the methods part of main text), we have the mapping $|0\rangle \mapsto |0000\rangle$, $|2\rangle \mapsto |0010\rangle$, $|6\rangle \mapsto |0110\rangle$. Thus we get the amplitude of the first-order coherence (like $|0000\rangle\langle 0010|$) and second-order coherence (like $|0000\rangle\langle 0110|$) as:

$$A_{1\text{st}} = \frac{-\tanh r}{\sqrt{2} \cosh r}, \quad A_{2\text{nd}} = \frac{-\sqrt{5} \tanh^3 r}{4 \cosh r},$$

where $r = \frac{1}{4} \ln \left(1 - 4 \frac{\xi}{\omega} \right)$. With the increasing of the squeezing parameter r , both $A_{1\text{st}}$ and $A_{2\text{nd}}$ grow, which are plotted in Figs. S7 c and d of the supplemental material. This leads to the enhancement of the zero-point fluctuation $\text{ZPF} = \sqrt{\langle \hat{x}^2 \rangle - \langle \hat{x} \rangle^2}$ with $\hat{x} = (\hat{a} + \hat{a}^\dagger)/2$ and $\hat{x}^2 = (\hat{a}^{\dagger 2} + \hat{a}^2 + \hat{a}^\dagger \hat{a} + \hat{a} \hat{a}^\dagger)/4$ (see Eqs. (S25)-(S28) in Sec. VI. of the supplemental material).

Therefore, here the enhancement effect refers to the final enhancement of ZPF of the squeezed vacuum states with increasing the antisqueezing strength. Finally, to measure the ZPF, we transfer the ZPF to the spin from the quantum fields by a SWAP gate. However, this is not the main result of this work, but an additional advantage of the introduced antisqueezing term.

In order to avoid the misleading, we have modified the inset of Fig. 2 and deleted the corresponding description in the abstract and added the pertinent discussion in Sec. VII of the revised supplementary material.

Comment 3-7

A minor comment: The coupling constants in figure 1c do not have the units.

Our response: In this revised manuscript, we have added the explanation on the units in caption of Fig. 1c.

Reviewers' Comments:

Reviewer #1:

Remarks to the Author:

As I mentioned in my previous report, I consider this a very interesting experimental work, which demonstrates the simulation of a spin-boson type model using NMR techniques. In the revised version of the manuscript, the authors now give a more accurate discussion about the equilibrium superradiant phase transition and the no-go theorem and make clear that their experiment only represent a simulation of the underlying Hamiltonians. This was my main point of critics in my previous report and I am happy with the revisions.

A remaining minor point concerns the discussion about the anti-squeezing term and the corresponding comment by Referee 2. I think the referee is right in saying that adding the anti-squeezing term is equivalent to reducing the prefactor of the A^2 term. So simulating both terms is in principle not necessary. The authors in turn insist in keeping the two terms separate since in the simulated physical model they have a different meaning and a different parameter-dependence. I think this point is still not very clear in the revised version. Therefore, I suggest that the authors revise once more the discussion in page 3, 1st column, where H_A is first introduced: E.g.,

"Note that formally, adding the H_{AS} simply renormalizes the prefactor of the A^2 term. However, in the simulated system the two contributions have a different physical meaning and a different parameter dependence Also in the experiments discussed below we will use different methods to account for the A^2 term and the anti-squeezing term ..."

or something along these lines. In any case it should be made clear that mathematically the A^2 term and the anti-squeezing term are the same.

Otherwise, I am happy with the revised version of the manuscript and after these minor corrections I recommend the paper for publication.

Reviewer #2:

Remarks to the Author:

I have read with interest the reply of the authors to my first report and to the reports from the other referees. The authors have addressed point by point all the raised issues and performed extensive modifications to the manuscript. Nevertheless, I am afraid that the response of the authors to my main concern is not convincing enough. In my previous report, I argued that the framing of the problem in terms of breaking a no-go theorem is misleading and that is not well supported by the reported experiment. I still think so. In particular, in my previous report, I had argued that the distinction between the A^2 term (in the cavity QED sense) and the anti-squeezing term was artificial because in the experiment only a term quadratic in the bosonic operators is implemented, which contains the sum of these two contributions. The authors argue that these two contributions are fundamentally different in two ways: (1) first, that their physical origin is different, as the first one arises, in principle, from the coupling of the atom to the EM field, while the second one needs to be engineered. (2) Second, that the dependence of each term in the atom-field coupling strength is different, which in turn leads to a modified behavior of the superradiant phase transition as a function of the coupling parameter (what the authors call a reversed-phase transition).

Regarding the first point (1), I would argue that this would be true in an actual cavity QED experiment, while in the presented simulation, both terms have the same origin. Moreover, their origin, here, is not physical as they are introduced a posteriori by a mathematical transformation. That is, the presented results for different values of the anti-squeezing term are all computed from the same experimental data corresponding to observables of the linear quantum Rabi model (without a quadratic term). Since the presence or not of a phase transition depends on the value of the anti-squeezing term and this is introduced in the post-processing of the data, one could argue that in the actual experiment a phase transition never takes place.

Regarding the second point (2), I agree that the dependence of each of the quadratic terms on the atom-field coupling strength is different, but how this modifies the critical behavior can be computed with a little bit of algebra and in no case constitutes a breaking of the no-go theorem. Let me explain. Given a quantum Rabi model with a linear and a quadratic term of strength λ and κ , respectively, the critical point is given by $\tilde{\lambda} = \sqrt{1 + 4\kappa/\omega}$, where $\tilde{\lambda}$ is defined as in (S2). The phase transition disappears if there is no real solution for λ (Condition). Now, if κ has only a contribution from the QED A^2 term, $\alpha \lambda^2/\Omega$, then the critical point is given by $\tilde{\lambda} = \sqrt{1 + \alpha \tilde{\lambda}^2}$, which does not have real solutions for $\alpha \geq 1$ (the no-go theorem). However, if we now introduce an additional quadratic contribution $-\xi$, such that $\kappa = \alpha \lambda^2/\Omega - \xi$, it becomes evident that real solutions of $\tilde{\lambda}$ exist for values of $\alpha > 1$, as long as $4\xi/\omega > 1$, and therefore a phase transition occurs. This is not in contradiction with the initial condition (Condition), and thus it does not constitute a breaking of the no-go theorem. The inversion of the behavior of the phase transition with the parameter $\tilde{\lambda}$ can be explained from the transformed parameters in (S4). As $\tilde{\lambda}$ is decreased, λ_s grows, and thus the behavior is reversed. This makes sense as for smaller coupling the anti-squeezing term prevails and the ground state tends towards a squeezed state. Nevertheless, the coupling $\tilde{\lambda}$ cannot be reduced arbitrarily as the Hamiltonian would become unbounded from below. This condition can be seen, for example, in the argument of the logarithm in (S2) that needs to remain positive, which restricts the values that $\tilde{\lambda}$ can take. This is a point that, I believe, the authors did not discuss.

Besides these arguments, I consider that the experiment is a beautiful quantum simulation of the quantum Rabi model and the manuscript contains an interesting discussion on the presence or not of a phase transition and the interplay between different Hamiltonian terms. In that sense, I think that, with the proper framing of the problem, the work deserves publication in a high-impact journal, although, in my view, it does not clear the bar of Nat. Commun.

Reviewer #3:

Remarks to the Author:

I am satisfied with the author's response. I think that all my questions were answered and the paper was improved. I would like to recommend the publication.

List of main changes

Main text:

1. We have added more discussion about the A^2 and anti-squeezing terms in the paragraph below Eq. (1).
2. Some experimental data with the implementation of squeezing operation are added into the inset of Fig. 3b from the supplemental material.
3. We have added some discussion about the experimental procedures with and without post-processing in the third subsection titled '*Recovering of SPT in the case of including A^2 term*' in Results (on Page 6).

Reply to Reviewer #1

Comment 1-1

As I mentioned in my previous report, I consider this a very interesting experimental work, which demonstrates the simulation of a spin-boson type model using NMR techniques. In the revised version of the manuscript, the authors now give a more accurate discussion about the equilibrium superradiant phase transition and the no-go theorem and make clear that their experiment only represent a simulation of the underlying Hamiltonians. This was my main point of critics in my previous report and I am happy with the revisions.

A remaining minor point concerns the discussion about the anti-squeezing term and the corresponding comment by Referee 2. I think the referee is right in saying that adding the anti-squeezing term is equivalent to reducing the prefactor of the A^2 term. So simulating both terms is in principle not necessary. The authors in turn insist in keeping the two terms separate since in the simulated physical model they have a different meaning and a different parameter-dependence. I think this point is still not very clear in the revised version. Therefore, I suggest that the authors revise once more the discussion in page 3, 1st column, where H_A is first introduced: E.g.,

"Note that formally, adding the H_{AS} simply renormalizes the prefactor of the A^2 term. However, in the simulated system the two contributions have a different physical meaning and a different parameter dependence Also in the experiments discussed below we will use different methods to account for the A^2 term and the anti-squeezing term ..."

or something along these lines. In any case it should be made clear that mathematically the A^2 term and the anti-squeezing term are the same.

Otherwise, I am happy with the revised version of the manuscript and after these minor corrections I recommend the paper for publication.

Our response: We really thank the referee for the detailed suggestion. In this revised version, we have added more discussion on Page 3, 1st column, according to the referee's suggestion.

The added text is "...Note that formally, adding the \hat{H}_{AS} simply renormalizes the prefactor of the A^2 term. However, in the simulated system, \hat{H}_A and \hat{H}_{AS} have a different physical meaning and a different parameter dependence on the changed parameter λ during checking the occurrence of SPT. In the following we will also discuss the different effects of A^2 term and the antisqueezing term on the SPT from the experimental results..."

Reply to Reviewer #2

Comment 2-1

.....Regarding the first point (1), I would argue that this would be true in an actual cavity QED experiment, while in the presented simulation, both terms have the same origin. Moreover, their origin, here, is not physical as they are introduced a posteriori by a mathematical transformation. That is, the presented results for different values of the anti-squeezing term are all computed from the same experimental data corresponding to observables of the linear quantum Rabi model (without a quadratic term). Since the presence or not of a phase transition depends on the value of the anti-squeezing term and this is introduced in the post-processing of the data, one could argue that in the actual experiment a phase transition never takes place.

Our response: In the manuscript, we actually state that, in the *simulated* cavity QED system, the A^2 term and antisqueezing term are physically different, not for the presented simulation. Also as suggested by Referee #1, we added a discussion to clarify this point again on Page 3, 1st column, where H_A is first introduced (see Reply to Comment 1-1).

Actually, we had also performed the corresponding simulation experiment of the QRM including antisqueezing term by physically implementing the antisqueezing operators (the experimental method shown in **Sec. Antisqueezing-enhanced ZPF** of the main text). The corresponding experimental data was shown in Fig. S3b of the supplemental material. In order to avoid the misleading again, in this revised version, we have also plotted the experimental data with physically implementing the antisqueezing term in the inset of Fig. 3b. From this figure, it is obvious that there is no qualitative difference between the cases with and without post-processing near the critical point. The order parameter in deep superradiant phase will drop down, because the truncated squeezing operator in small Hilbert space with large squeezing parameter r will loss the validity (see Fig. S4). In order to demonstrate the occurrence of the SPT in a more effective way with the limited qubits, in the main text, we have also performed an alternative experimental scheme by means of the post-processing. Based on the experimental data with and without post-processing for the antisqueezing term, we can state that our experiments indeed demonstrate that the antisqueezing effect could recover the SPT in the system of including A^2 term with $\alpha > 1$. The corresponding discussion is also added into the 1st and 2st paragraphs of page 5.

We have added a paragraph of discussion on Page 6,

"...Actually, we had also performed the corresponding simulation experiment without post-processing by physically implementing the squeezing operation $\hat{S}(\tilde{r})$ in different parameter regions, and obtained the order parameter Φ by experimentally preparing the ground state of the original Hamiltonian \hat{H} . The corresponding experimental results are also plotted in the inset of Fig. 3b as well as in Supplementary Figure 3a [47]. It is clearly seen that there is a similar rapid growth of the order parameter Φ near the critical point, indicating the occurrence of SPT, for both the cases with and without post-processing. The order parameter Φ without post-processing in deep superradiant phase will drop down because the truncated squeezing operator in small-size Hilbert space with large squeezing parameter r will loss the validity (see Supplementary Figure 4). The method of the post-processing avoids this limit of the restricted Hilbert space in realizing $\hat{S}(\tilde{r})$..."

Comment 2-2

Regarding the second point (2), I agree that the dependence of each of the quadratic terms on the atom-field coupling strength is different, but how this modifies the critical behavior can be computed with a little bit of algebra and in no case constitutes a breaking of the no-go theorem. Let me explain. Given a quantum Rabi model with a linear and a quadratic term of strength λ and κ , respectively, the critical point is given by $\tilde{\lambda} = \sqrt{1 + 4\kappa/\omega}$, where $\tilde{\lambda}$ is defined as in (S2). The phase transition disappears if there is no real solution for λ (Condition). Now, if κ has only a contribution from the QED A^2 term, $\alpha\lambda^2/\Omega$, then the critical point is given by $\tilde{\lambda} = \sqrt{1 + \alpha\tilde{\lambda}^2}$, which does not have real solutions for $\alpha \geq 1$ (the no-go theorem). However, if we now introduce an additional quadratic contribution $-\xi$, such that $\kappa = \alpha\lambda^2/\Omega - \xi$, it becomes evident that real solutions of $\tilde{\lambda}$ exist for values of $\alpha > 1$, as long as $4\xi/\omega > 1$, and therefore a phase transition occurs. This is not in contradiction with the initial condition (Condition), and thus it does not constitute a breaking of the no-go theorem.

Our response: By analytically calculating the influence of the A^2 and antisqueezing terms on the critical behavior, this comment (similar as the discussion shown in Section I of our supplemental material) mathematically demonstrates that the introduced antisqueezing term could recover the SPT in the system with A^2 term and $\alpha > 1$. The *Condition* called here actually is a general *mathematical* method for identifying the occurrence of phase transition, which is also applied in our work. However, as shown in the above reply, we did not only introduce the influence of A^2 and antisqueezing term on the SPT via the well-defined squeezing transformation. Actually, we had done the corresponding experiment for the system including anti-squeezing term, which experimentally demonstrates that the anti-squeezing effect could recover the SPT in the case of including A^2 term with $\alpha > 1$. In this sense, we state that our experiment simulates the SPT beyond no-go theorem via antisqueezing.

Here we would like to clarify our statement more clearly to avoid misleading again. Our work actually is not in contradiction with the mathematical *Condition* of checking SPT in the no-go theorem, and we also did not want to demonstrate the original no-go theorem is incorrect. Besides the mathematical description, the no-go theorem has clearly physical meaning. As shown in Ref. [14], it is demonstrated that the A^2 term with $\alpha > 1$ should be included into the cavity QED system when the atom-field interaction is correctly described. In this case, one could argue that the expected SPT in the Dicke model will disappear in the actual cavity QED. From a physical point of view, the original definition of no-go theorem describes a physical phenomenon, i.e., the SPT disappears in the cavity QED system when A^2 term with $\alpha > 1$ is included correctly. By using quantum simulation techniques, here we experimentally demonstrate that the disappeared SPT could be recovered even in the case of including A^2 term with $\alpha > 1$, if one could introduce a proper antisqueezing term into the actual system. Our experiment is not in contradiction with the original no-go theorem, and it indeed relaxes the limit of A^2 term on SPT via introducing the antisqueezing effect. In this sense, our work demonstrates the SPT beyond no-go theorem via antisqueezing.

To avoid possible misunderstanding again, the above discussion has been properly added in this revised manuscript, and we have deleted the words "break the no-go theorem" in the whole manuscript.

In Discussion part, the modified text is "*In summary, from a physical point of view, we have presented a first proof-of-principle experiment to demonstrate equilibrium SPT beyond no-go theorem induced by the antisqueezing effect. Our experiment is not in contradiction with the original no-go theorem, and it indeed relaxes the limit of A^2 term on SPT via introducing the*

antisqueezing effects...."

Comment 2-3

The inversion of the behavior of the phase transition with the parameter $\tilde{\lambda}$ can be explained from the transformed parameters in (S4). As $\tilde{\lambda}$ is decreased, λ_s grows, and thus the behavior is reversed. This makes sense as for smaller coupling the anti-squeezing term prevails and the ground state tends towards a squeezed state. Nevertheless, the coupling $\tilde{\lambda}$ cannot be reduced arbitrarily as the Hamiltonian would become unbounded from below. This condition can be seen, for example, in the argument of the logarithm in (S2) that needs to remain positive, which restricts the values that $\tilde{\lambda}$ can take. This is a point that, I believe, the authors did not discuss.

Our response: As shown in this comment, the coupling $\tilde{\lambda}$ cannot be reduced arbitrarily as the Hamiltonian would become unbounded (i.e., system becomes unstable) from below. As shown in Figs. 3b and 4b, there exists unstable phase (UP) in our model, and more detailed analysis was described in the supplementary material (paragraph below equation S15). According to the referee's suggestion, we have added the analytical expression of system becoming unbounded together with more discussion on page 6 above the Discussion of the main text.

The added text is *"...However, too large antisqueezing strength will bring the system into an unstable phase (UP), when the rescaled ground-state excitation becomes an imaginary number, i.e., $1 + \alpha\tilde{\lambda}^2 - 4\xi/\omega < 0$, corresponding to $\tilde{\lambda} < \sqrt{(1/\alpha)(4\xi/\omega - 1)}$. In other words, the coupling $\tilde{\lambda}$ cannot be reduced arbitrarily as the Hamiltonian would become unbounded from below..."*

Comment 2-4

Besides these arguments, I consider that the experiment is a beautiful quantum simulation of the quantum Rabi model and the manuscript contains an interesting discussion on the presence or not of a phase transition and the interplay between different Hamiltonian terms. In that sense, I think that, with the proper framing of the problem, the work deserves publication in a high-impact journal, although, in my view, it does not clear the bar of Nat. Commun.

Our response: We thank the referee for the positive comment. After the above modification, the previous misleading has been clarified. We believe that the manuscript is now suitable for publication in Nat. Commun.

Reply to Reviewer #3

Comment 3-1

I am satisfied with the author's response. I think that all my questions were answered and the paper was improved. I would like to recommend the publication.

Our response: We thank the referee's recommendation.

Reviewers' Comments:

Reviewer #2:

Remarks to the Author:

I have read the reply of the authors to my previous report and I think that the performed modifications are sufficient to clarify the main concerns that I had raised. In particular, the equivalence between the A^2 term and the squeezing operation is now more clear, and the notion that this experiment does not constitute a counterexample to the no-go theorem of superradiance is now stated.

While I still would not consider the reported experiment proof of the existence or not of a superradiant phase transition, I do believe that the work deserves publication as it constitutes an impressive demonstration of the state-of-the-art in quantum simulation with nuclear spins. And in that spirit, I recommend its publication.

Reply to Reviewer #2

I have read the reply of the authors to my previous report and I think that the performed modifications are sufficient to clarify the main concerns that I had raised. In particular, the equivalence between the A^2 term and the squeezing operation is now more clear, and the notion that this experiment does not constitute a counterexample to the no-go theorem of superradiance is now stated.

While I still would not consider the reported experiment proof of the existence or not of a superradiant phase transition, I do believe that the work deserves publication as it constitutes an impressive demonstration of the state-of-the-art in quantum simulation with nuclear spins. And in that spirit, I recommend its publication.

Our response: We really thank the reviewer's recommendation. His/her suggestions help us a lot to improve the quality of the manuscript.